# An optimized method for 3D fluorescence co-localization applied to human kinetochore protein architecture

**Aussie Suzuki\*, Sarah K Long, Edward D Salmon**

Department of Biology, University of North Carolina at Chapel Hill, Chapel Hill, United States

**Abstract** Two-color fluorescence co-localization in 3D (three-dimension) has the potential to achieve accurate measurements at the nanometer length scale. Here, we optimized a 3D fluorescence co-localization method that uses mean values for chromatic aberration correction to yield the mean separation with ~10 nm accuracy between green and red fluorescently labeled protein epitopes within single human kinetochores. Accuracy depended critically on achieving small standard deviations in fluorescence centroid determination, chromatic aberration across the measurement field, and coverslip thickness. Computer simulations showed that large standard deviations in these parameters significantly increase 3D measurements from their true values. Our 3D results show that at metaphase, the protein linkage between CENP-A within the inner kinetochore and the microtubule-binding domain of the Ndc80 complex within the outer kinetochore is on average ~90 nm. The Ndc80 complex appears fully extended at metaphase and exhibits the same subunit structure in vivo as found in vitro by crystallography.
DOI: https://doi.org/10.7554/eLife.32418.001

**\*For correspondence:**
suzukia@email.unc.edu

**Competing interests:** The authors declare that no competing interests exist.

## Introduction

Confocal light microscopy (LM) is widely used for understanding protein localization, architecture, and functions in cells, but the maximum resolution without any image processing is still only ~200 nm in the x-y image plane and ~400 nm along the z-axis. The scale of many protein complexes is smaller than the diffraction limit of LM. In order to deduce the function of protein complexes, it is critical to develop a method to determine the three-dimensional (3D) locations of proteins within a complex in their native environment.

Kinetochores are protein assemblies at the periphery of centromeric chromatin that have vital functions for achieving accurate chromosome segregation during mitosis and meiosis. A modified histone H3, CENP-A, within nucleosomes at centromeric chromatin marks the site of kinetochore assembly. CENP-A recruits ~25 different kinds of core-structural proteins including Constitutive Centromere Associated Protein Network (CCAN) proteins and outer kinetochore proteins called the KMN network (Knl1, Mis12 complex, and Ndc80 complex) (*Cheeseman et al., 2006*; *McKinley and Cheeseman, 2016*; *Musacchio and Desai, 2017*; *Pesenti et al., 2016*; *Takeuchi and Fukagawa, 2012*). Two important linkers in human kinetochores are CENP-C and CENP-T because they connect between centromeric chromatin and the KMN network proteins (*Huis In 't Veld et al., 2016*; *Klare et al., 2015*; *Musacchio and Desai, 2017*; *Nishino et al., 2013*; *Nishino et al., 2012*; *Screpanti et al., 2011*; *Suzuki et al., 2015*; *Suzuki et al., 2014*). The highly conserved Ndc80 complex (Hec1(Ndc80), Nuf2, Spc24, Spc25) is one of the best structurally characterized kinetochore proteins, and it has a major role in kinetochore microtubule (kMT) attachment (*Ciferri et al., 2008*; *Musacchio and Desai, 2017*).

We previously reported a nm-scale map of how human kinetochore proteins are organized on average along the inner to outer kinetochore axis at metaphase, obtained using the 'Delta' two-color fluorescence co-localization method (*Wan et al., 2009*). The Delta method provided local chromatic aberration (CA) correction in the x-y image plane by measurements along the K-K axis between pairs of sister kinetochores. More recently, *Fuller and Straight (2012)* developed a 3D method (termed CICADA [Co-localization and In-situ Correction of Aberration for Distance Analysis]) to measure the mean separation of red and green fluorescent centroids at individual kinetochores while locally correcting for CA in the x, y, and z directions using a third color (*Fuller and Straight, 2012*). Their method yielded an identical value to what was obtained by the Delta method (*Fuller and Straight, 2012*; *Suzuki et al., 2014*; *Wan et al., 2009*). In contrast, a recent report by *Smith et al., 2016* used a 3D measurement method to obtain mean separation values for protein domains within kinetochores that were significantly greater than our Delta values and the 3D value obtained by the CICADA method (*Fuller and Straight, 2012*; *Smith et al., 2016*; *Suzuki et al., 2014*; *Wan et al., 2009*). This group questioned the accuracy and significance of the Delta values because they were '1D measurements and neglected z-axis information'.

In this paper, we have tested the above conclusion by optimizing a 2D and 3D fluorescence co-localization method to obtain the mean separation between the centroids of green and red foci with ~10 nm accuracy (*Figure 1*). The method depends on correcting CA based on mean values from a fluorescent standard, a method commonly used for fluorescence co-localization, but not previously optimized and rigorously analyzed for sources of error, as performed here. Computer simulations were used to understand how well the measurement method works and to find how accuracy is limited in our method. This study revealed major sources of error produced by standard deviations (SDs) in fluorescent centroid determination, CA correction, and coverslip thickness, as well as addressing several issues concerning fixation methods. We found that our 3D measurements applied to human kinetochores were similar to the Delta measurements and much shorter than those reported by *Smith et al., 2016*. Their larger measurements may be explained by much larger SDs in centroid determination and CA correction than typical of our experiments. We also used the known structure of the Ndc80 complex as a 'Molecular Ruler' to verify the accuracy of our measurement method, and to determine its subunit structure and conformation within the kinetochore.

## Results

### Principles of the method

#### Sample preparation

HeLa cells were grown on #1.5 coverslips, the design value for the microscope objective, to prevent Z-axis CA (chromatic aberration) (*Figure 1A*). Cells were fixed with paraformaldehyde (PFA) without pre-extraction or supplement of detergent in the PFA solution, and stained with primary antibodies using a protocol from previous studies (*Figure 1A*) (*Suzuki et al., 2015*; *Suzuki et al., 2014*; *Suzuki et al., 2011*). The concentration of green and red secondary antibodies was adjusted to give nearly equal fluorescence intensity (*Figure 1B*). Integrated kinetochore fluorescence above background (>100,000 photon counts per kinetochore) was important to minimize the SD of centroid determination (CDsd) close to 5 nm (*Thompson et al., 2002*). Labeled samples were typically mounted with ProLong Gold mounting media. For the cellular CA correction standard, a monoclonal antibody (9G3) was used to label Hec1 (hsNdc80) protein at a site just inside of its MT binding CH domain, and this antibody was double labeled with green and red fluorescence by secondary antibodies (called Hec1-9G3-GR).

#### Imaging

Imaging was performed using a confocal light microscope with a high NA objective lens (100x/1.4 NA) and a camera with small pixel size (64 nm, less than 1/3 resolution limit) (*Figure 1C*). Metaphase cells were positioned in the central field of view of the camera. Images were recorded at 200 nm intervals along the z-axis beginning just before the coverslip surface and extending to the top of the spindle to measure equal numbers of kinetochores below and above the middle of the spindle.

### A. *Fixation

*Use #1.5 coverslip
(0.17 mm thickness)

3% PFA (37°C)

No detergent before
or in fixation solution.
It causes shrinkage of
cellular strucutre.

### B. Staining

Make *high S/N samples

This is a critical factor
for accuracy in
determination of
fluorescent centroid.

### C. Imaging

Mount fixed and stained samples
with *Prolong Gold

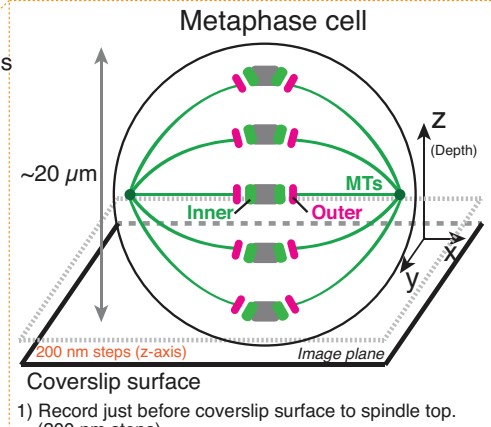

Metaphase cell

~20 μm

z (Depth)

MTs

Inner    Outer

200 nm steps (z-axis)    Image plane

Coverslip surface

1) Record just before coverslip surface to spindle top.
   (200 nm steps)
2) Sequentially take Red and Green images at same
   image plane.

### D. *CA correction

Use Hec1-9G3-GR for
CA correction

Hec1-9G3-GR

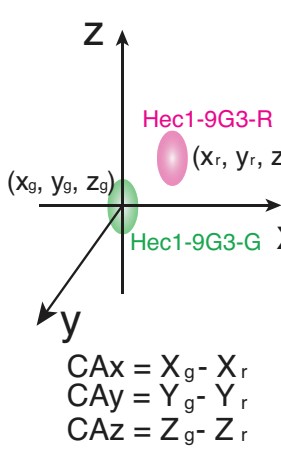

Hec1-9G3-R
$(x_r, y_r, z_r)$

$(x_g, y_g, z_g)$

Hec1-9G3-G

$CAx = X_g - X_r$
$CAy = Y_g - Y_r$
$CAz = Z_g - Z_r$

### E. Measurements (Delta, single kinetochore 2D and 3D)

Measure sister kinetochore pairs 1) to compare directly between
Delta and 2D/3D Fluorescent Co-localization Analysis using the same data sets; and
2) to obtain 2D and 3D angles relative to K-K axis.

**Delta (1D)
(Sister kinetochore)
No CA correction required.**

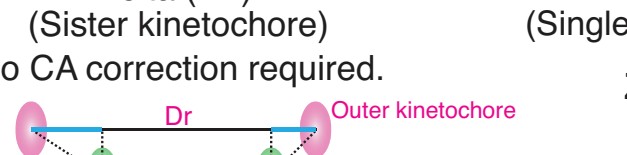

Dr    Outer kinetochore

Dg    Inner kinetochore

$Delta = (D_r - D_g)/2$

**2D, 3D
(Single kinetochore)**

z

Sz

Outer kinetochore
$(x_r, y_r, z_r)$

$(x_g, y_g, z_g)$

S(3D)

Inner kinetochore    Sx

Sy    S(2D)

X (K-K axis)

y

$Sx = X_g - X_r - meanCA_x$
$Sy = Y_g - Y_r - meanCA_y$
$Sz = Z_g - Z_r - meanCA_z$

**Separation**

$S(2D) = mean((Sx^2 + Sy^2)^{0.5})$
$S(3D) = mean((Sx^2 + Sy^2 + Sz^2)^{0.5})$

### F. Error analysis

Measure sources of experimental variance and use computer simulations
to evaluate their contribution to experimental measurements. Sources
include embedding media, coverslip thickness, kinetochore tilt, fixation
methods, SD in centroid determination (CDsd), and SD in CA correction
(CAsd).

**Figure 1.** Principle of procedure used in this study. (**A**) Fixation: 3% PFA in PHEM buffer without any detergent was used as standard fixation in this study (Details in Text and Methods). (**B**) Staining: Fixed samples were immunofluorescently labeled with antibodies to enhance S/N. Samples were usually mounted with Prolong Gold. (**C**) Imaging was performed by spinning disk confocal microscopy (See Materials and methods for details). (**D**) For CA correction, Hec1-9G3 monoclonal antibody was labeled by green and red tagged secondary antibodies (Hec1-9G3-GR). (**E**) Measurements: X, Y, Z coordinates of the centroids of green and red kinetochore fluorescence were determined by 3D Gaussian fitting. Details of equations for separation measurements are in Text, Figures, and Methods. (**F**) Error analysis: computer simulations were used for error analysis of the method. The sources of potential errors in separation measurements were highlighted in pink underline with *.

*Figure 1 continued on next page*

*Figure 1 continued*

DOI: https://doi.org/10.7554/eLife.32418.002

The following source data is available for figure 1:

**Source data 1.** Pipeline for obtaining mean 3D separation measurements with nm-scale accuracy.

DOI: https://doi.org/10.7554/eLife.32418.003

## Image analysis

We selected sister kinetochores for measurement not only for Delta values (*Figure 1D–E*), but also to obtain the direction of the sister-sister kinetochore axis (K-K). For each kinetochore, we obtained the x, y, z coordinates for the centroids of green (Xg, Yg, Zg) and red (Xr, Yr, Zr) using 3D Gaussian fitting methods (*Wan et al., 2009*). The Sx, Sy, and Sz coordinates of the separation vector S between green and red foci were calculated from Sx = Xg-Xr-meanCAx (CAmx), Sy = Yg-Yr-mean-CAy (CAmy), and Sz = Zg-Zr-meanCAz (CAmz) (*Figure 1E*), where the mean values of CAx, CAy, and CAz were obtained from kinetochores labeled with Hec1-9G3-GR within our measurement volume (*Figure 1D*). 'Raw' values for the mean separation vector lengths in 2D and 3D were calculated for each pair of green-red kinetochore labels as $S(2D) = \text{mean} ((Sx^2 + Sy^2)^{0.5})$ and $S(3D) = \text{mean} ((Sx2 + Sy2 + Sz2)^{0.5})$ from measurements of 200 to 400 kinetochores obtained from 5 to 20 different cells (*Figure 1E*). The mean of these 'raw' separation values (S(2D), and S(3D)) were over-estimates (*Churchman et al., 2006*; *Churchman et al., 2005*). In order to obtain more accurate mean separation values, a maximum likelihood method was applied to the raw separation data (called ML2D and ML3D) (*Churchman et al., 2006*; *Churchman et al., 2005*).

## Error analysis

Potential errors from different embedding media, the wrong coverslip thickness, kinetochore tilt, and different fixation methods were examined experimentally (Listed in *Figure 1—source data 1*). A computer simulation of the measurement method was used to evaluate the significance of these errors, as well as errors produced by the SDs in centroid determination (CDsd) and CA (CAsd) over the field of measurement (*Figure 1F*).

## Chromatic aberration (CA) measurements from a cellular standard

CA within our measurement was determined from CAx = Xg-Xr, CAy = Yg-Yr, and CAz = Zg-Zr measured for Hec1-9G3-GR labeled kinetochores of metaphase cells (*Figures 1D* and *2*). The mean values of CAx, CAy, and CAz plus their standard deviations (SDs) are listed in *Table 1* for several different preparations. Note the mean values for CAx, CAy, and CAz were each very similar between different preparations. We included one preparation where the embedding solution was PBS. PBS has a refractive index near water, 1.33, and far from the refractive index of the coverslip (~1.52). The mean CAx, CAy, and CAz values were similar to those using the ProLong Gold embedding medium (*Table 1*). The major difference was a much larger SD for CAz compared to ProLong Gold (48 nm vs ~18 nm, *Table 1*). This occurs because the lower refractive index induces spherical aberration that increases with z-axis depth from the coverslip surface (*Figure 2—figure supplement 1*). We also tested one sample with glutaraldehyde (GA) fixation and embedding with ProLong Gold. The mean CA values were nearly identical to the results obtained from PFA fixation (*Table 1*).

A small measurement region in the center of the camera detector (~300×200 pixels) was used for the entire analysis in this study (*Figure 2—figure supplement 2A–B*). In the measurement region, there was a gradient for CAx in the x direction (~30 nm/ 280 pixels (19 µm)), but not in the y or z directions. There was a gradient for CAy in the y direction (~21 nm/180 pixels (11.5 µm)), but not in the x or z directions. There was no noticeable gradient for CAz in either the x, y, or z directions compared to the SD (*Figure 2—figure supplement 3*). These gradients in CA had no more effect on measurement accuracy than randomly distributed CA values with the same mean and SD (see computer simulations below).

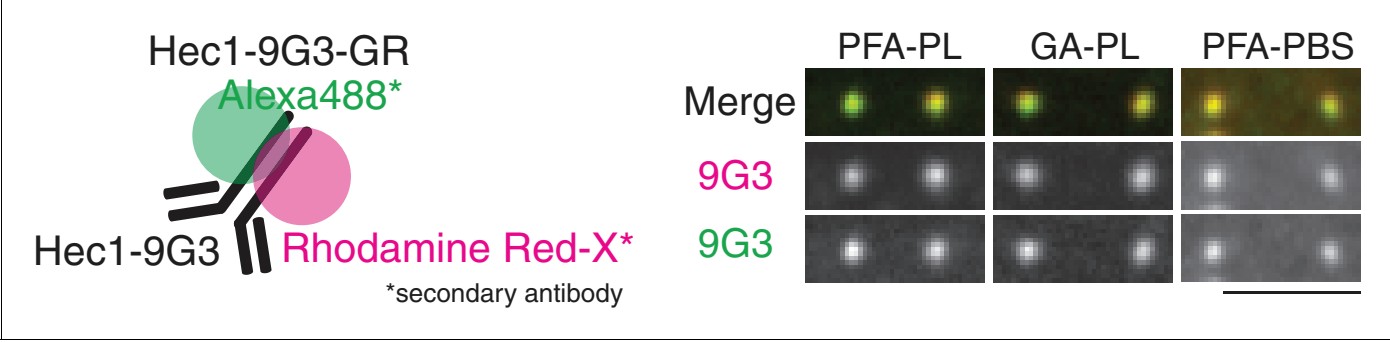

**Figure 2.** CA (Chromatic aberration) measurements using Hec1-9G3-GR. Schematic model of Hec1-9G3-GR. Hec1-9G3 antibody was labeled by secondary antibodies with Rhodamine Red-X and Alexa 488 (left). Representative Hec1-9G3-GR images in samples fixed by PFA or GA mounted with Prolong Gold or PBS.

DOI: https://doi.org/10.7554/eLife.32418.004

The following figure supplements are available for figure 2:

**Figure supplement 1.** CAz is insensitive to depth in samples mounted with Prolong Gold, but not in samples mounted with PBS.

DOI: https://doi.org/10.7554/eLife.32418.005

**Figure supplement 2.** Center of camera image was used for all analysis.

DOI: https://doi.org/10.7554/eLife.32418.006

**Figure supplement 3.** Measurement of CA of the microscope optics.

DOI: https://doi.org/10.7554/eLife.32418.007

## Validation of our 2D and 3D co-localization method for known separations within the kinetochore

We initially measured the 2D and 3D separations between an antibody to CENP-T (CENP-T (M), *Gascoigne et al., 2011*) and Hec1-9G3 because previous studies reported values of Delta and 3D mean separations (by the CICADA method) using these same antibodies (*Fuller and Straight, 2012*; *Suzuki et al., 2014*; *Wan et al., 2009*). The reported 1D and 3D values for CENP-T (M) and Hec1-9G3 were ~60 and ~62 nm respectively. Our mean separation ±SD between CENP-T (M) and Hec1-9G3 for Delta = 60 ± 7.5 nm, S(2D)=63 ± 11.2 nm, ML2D = 62 ± 11.3 nm, S(3D)=69 ± 12.3 nm, and ML3D = 66 ± 12.5 nm (*Figure 3A–B*), almost identical to the previous studies (*Fuller and Straight, 2012*; *Suzuki et al., 2014*; *Wan et al., 2009*).

The orientation of the separation vector at each kinetochore was also calculated relative to the K-K axis. For CENP-T(M) to Hec1-9G3, the mean vector angle to the K-K axis in the X-Y plane, angle α = −0.3 ± 20.1 degrees, while the mean vector angle relative to the Z-axis (angle β) was −1.8 ± 23.3 degrees (*Figure 3C*). A significant fraction of the SD in these angles may be due to errors from CDsd and CAsd (see computer simulations below).

To further test the accuracy of our 2D/3D co-localization method, we examined the mean separation of different domains within the Ndc80 complex. We chose the Ndc80 complex as a molecular ruler, since its protein structure is well known. The Ndc80 complex contains a dimer of Ndc80/Nuf2 and a dimer of Spc24/Spc25 (*Cheeseman et al., 2006*; *Ciferri et al., 2008*; *Musacchio and Desai,*

**Table 1.** Mean and SD values of CAx, CAy, CAz for different sample preparations.

|  | Fixation | MM | Green | Red | CAx (nm) | CAy (nm) | CAz (nm) | N |
|---|---|---|---|---|---|---|---|---|
| Use for correction | PFA | PL | 9G3 | 9G3 | 13.1 ± 9.1 | 15.8 ± 7.5 | 3.5 ± 17.6 | 166 |
|  | PFA | PL | 9G3 | 9G3 | 14.7 ± 11.4 | 13.2 ± 12.1 | 1.5 ± 17.0 | 256 |
|  | PFA | PL | 9G3 | 9G3 | 15.7 ± 12.5 | 15.6 ± 8.2 | −4.8 ± 20.2 | 206 |
| Use for correction | PFA | PBS | 9G3 | 9G3 | 13.2 ± 10.7 | 18.7 ± 10.4 | −2.7 ± 47.5 | 168 |
| Use for correction | GA | PL | 9G3 | 9G3 | 19.2 ± 10.3 | 16.0 ± 9.0 | −6.8 ± 18.8 | 170 |

DOI: https://doi.org/10.7554/eLife.32418.008

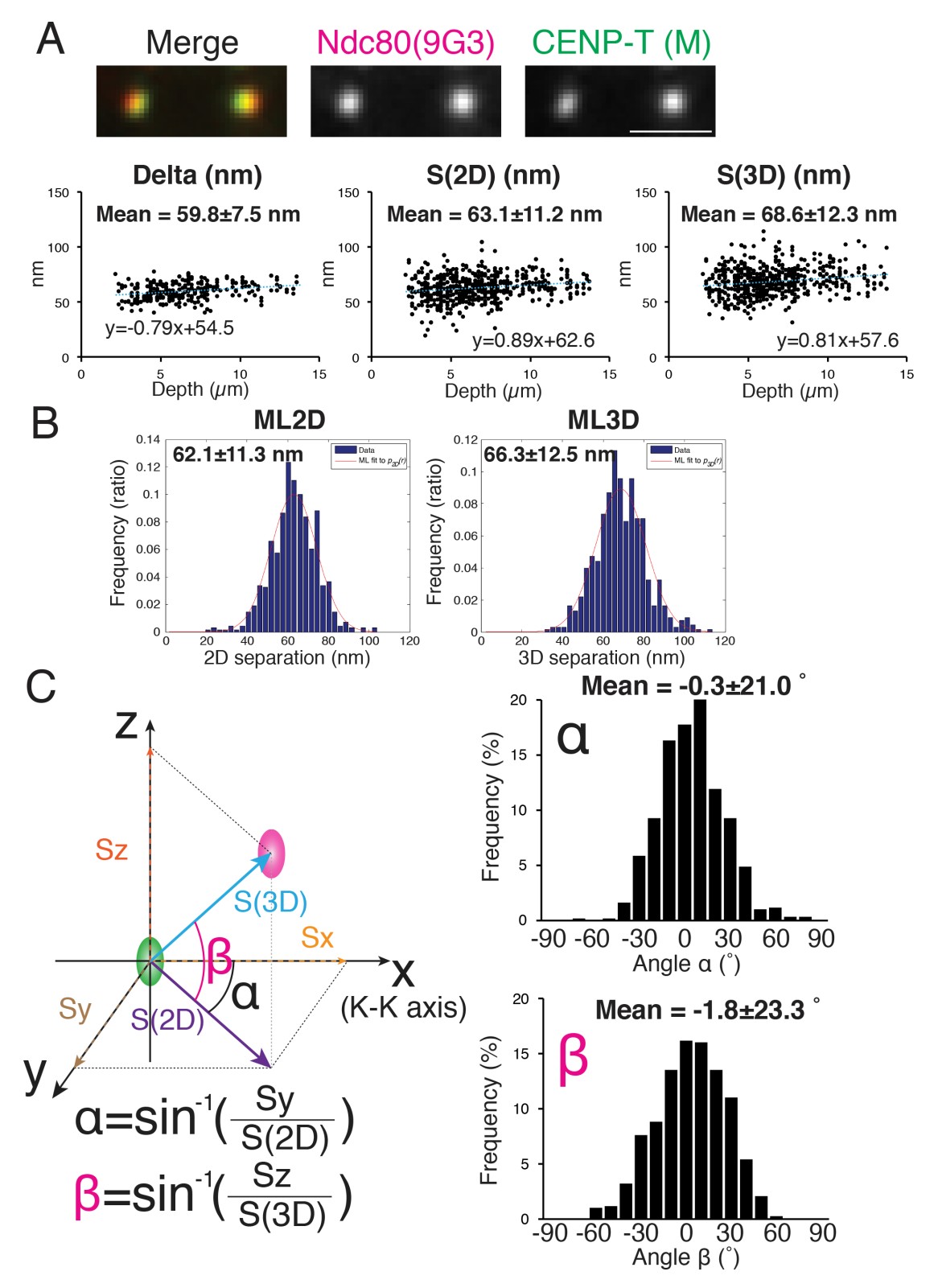

**Figure 3.** Mean Delta, 2D, and 3D separations between Hec1-9G3 and CENP-T (M). (**A**) Representative immunofluorescent (IF) images (top) and plots of Depth verses Delta, S(2D) and S(3D) for the separation between CENP-T(M) and Hec1-9G3 (bottom). (**B**) Histograms of maximum likelihood fit used to calculate ML2D and ML3D. (**C**) Orientation of the S(3D) separation vector relative to the sister K-K axis (x-axis in this diagram). Angle α is the rotation of the S(2D) vector component in the x-y plane away from the K-K (x-axis) and angle β is the rotation of the S(3D) vector away from the x-y plane and

*Figure 3 continued on next page*

*Figure 3 continued*

toward the z-axis (schematic on left). Histograms of α and β values (Right). n > 600 individual kinetochores. All values were mean ±SD. Note, CENP-T(M) is a polyclonal antibody against whole CENP-T protein, and its epitope is not precisely known (*Gascoigne et al., 2011*).

DOI: https://doi.org/10.7554/eLife.32418.009

*2017*). Most of the length of the Ndc80 complex is alpha helical coiled-coil, and its contour length is 55 ~ 60 nm, as measured in vitro by electron microscopy and crystallography (*Ciferri et al., 2008*; *Huis In 't Veld et al., 2016*; *Valverde et al., 2016*; *Wei et al., 2005*). To reveal the cellular Ndc80 complex structure, we developed HeLa cells stably expressing EGFP-Ndc80, Ndc80-EGFP, EGFP-Nuf2, Nuf2-EGFP, EGFP-Spc24, Spc24-EGFP, EGFP-Spc25, or Spc25-EGFP (*Figure 4A*). *Figure 4B*

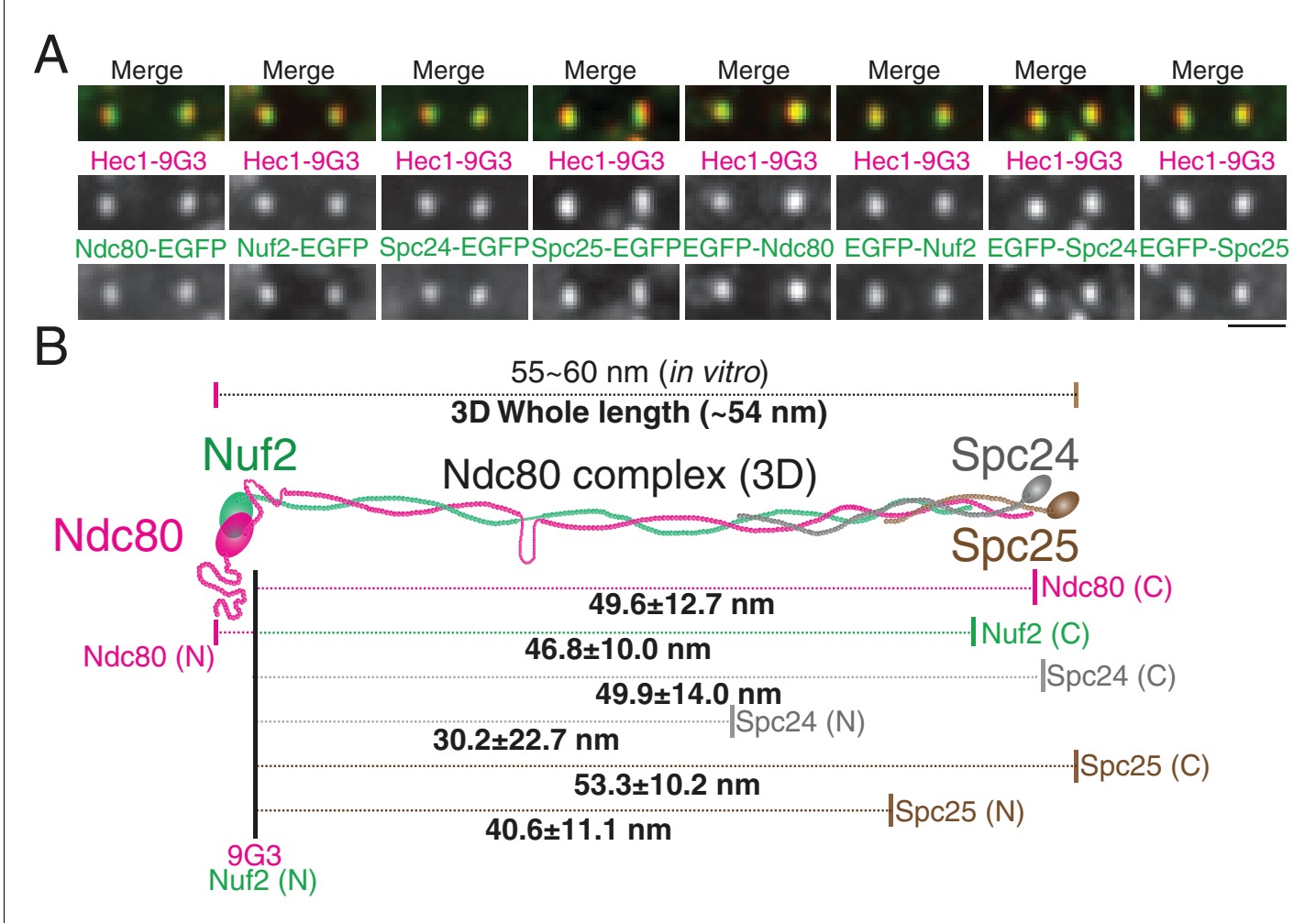

**Figure 4.** 2D/3D co-localization measurements of a structural standard, the extended Ndc80 complex. (**A**) Representative immunofluorescent images of Hec1-9G3 and EGFP fused to different protein domains along the Ndc80 complex (See Text and Materials and methods). (**B**) Schematic summary of the ML3D Ndc80 complex measurements at metaphase. All values were mean ±SD. n > 200 kinetochores for each measurement. Detail values are listed in *Figure 4—figure supplement 1*.

DOI: https://doi.org/10.7554/eLife.32418.010

The following source data and figure supplement are available for figure 4:

**Source data 1.** The list for Ndc80 complex measurements.

DOI: https://doi.org/10.7554/eLife.32418.012

**Figure supplement 1.** Detailed sub-protein structure of Ndc80 complex.

DOI: https://doi.org/10.7554/eLife.32418.011

and *Figure 4—source data 1* show the separation values relative to Hec1-9G3 to different ends of the Ndc80, Nuf2, Spc24, and Spc25 proteins, tagged in each case with EGFP. For example, the mean separation measurements between Spc25-EGFP and Hec1-9G3 were Delta = 49 ± 5 nm, S(2D) =52 ± 10 nm, S(3D)=55 ± 10 nm, ML2D = 51 ± 10 nm, and ML3D = 53 ± 10 nm. The Ndc80 complex is likely extended along kMTs since the Delta and ML3D values for the length of the Ndc80 complex are nearly identical. Also, overall, our Delta, 2D, and 3D mean separation measurements yielded a close fit to a Ndc80 complex structure that was determined in vitro (*Ciferri et al., 2008*; *Valverde et al., 2016*). As expected, the C-terminus of Spc24 is located a few nm inside of the C-terminus of Spc25, and Spc24 is longer than Spc25 (*Figure 4—figure supplement 1*) (*Valverde et al., 2016*). The CH-domain of Nuf2 is located a few nm inside of the N-terminal CH domain of Ndc80, and Ndc80 is longer than Nuf2 (*Figure 4—figure supplement 1*) (*Ciferri et al., 2008*; *Valverde et al., 2016*). Unexpectedly, in vivo, the C-terminus of Ndc80 locates much closer to the globular domains of Spc24/Spc25 (*Figure 4B*). This difference might be caused by structural restrictions on the position of the EGFP tag, which was tethered to the protein end by a nine amino acid linker.

Sister kinetochore pairs were used for the majority of measurements in this study to directly compare separations obtained by Delta and 2D/3D fluorescent co-localization methods using the same data sets. To test if selection of sister pairs influenced our measurements, we measured 120 kinetochores selected randomly in a single metaphase cell. This produced a mean separation value nearly identical to the mean value for single kinetochore measurements of sister pairs obtained from six metaphase cells (*Table 2*).

## The mean human kinetochore protein architecture at metaphase measured by the Delta method and our 3D method are similar

*Figure 5* shows how the centroids of protein labels are separated on average along the inner-outer kinetochore axis based on our 2D/Delta measurements (Top) and 3D measurements (bottom). All of our 2D and Delta values are nearly identical (usually less than 3 nm difference). In general, our 3D measurements were only ~5 nm larger than the Delta or 2D values (*Figure 5* and the *Figure 5— source data 1*). Unlike the 3D kinetochore protein architecture proposed by (*Smith et al., 2016*), our data showed no ~70 nm gap between CCAN protein domains and the Ndc80 complex. The inner kinetochore proteins are much closer to the outer kinetochore, corresponding to the previous studies using Delta measurements (*Suzuki et al., 2014*). For example, our 3D mean separation between CENP-A and Hec1-9G3 is ~90 nm, which is nearly identical to values obtained by Delta (~84 nm) and ML2D (~85 nm). In *Figure 5*, the Ndc80 complex and protein linkage to the inner kinetochore are shown aligned with the axis of kMTs because the ML3D mean separation values are only slightly greater than their corresponding Delta values, and the Delta value is a projection of the ML3D value along the K-K axis.

## Coverslip thickness both greater and less than 0.17 mm produces z-axis chromatic aberration (Z-offset) independent of focal depth

Normally the value of Z-offset (mean Sz) for each coverslip-slide preparation was within ±25 nm of zero (*Figure 5—source data 1*). However, occasionally, significantly large positive or negative values occurred, which made the 3D measurement much larger than for small Z-offsets. The largest Z-offset occurred for one sample stained for CENP-T(M) and 9G3,~80 nm (*Table 3A*), which increased the S (3D) from 65 to 103 nm (*Table 3A*,). If this large Z-offset was subtracted from the Sz data, the mean zS3D (S(3D) with subtracted Z-offset) value became close to the value with near zero Z-offset,~65 nm (*Table 3A*). This comparison indicated that the higher Z-offset was independent of the actual

**Table 2.** Comparison of measurements for kinetochores that were part of sister pairs and kinetochores selected at random from a single metaphase cell.

| Methods | Green | Red | Delta (nm) | S(2D) (nm) | S(3D) (nm) | ML2D (nm) | ML3D (nm) | n |
|---|---|---|---|---|---|---|---|---|
| Sister pair | Spc25-EGFP | 9G3 | 49.4 ± 4.6 | 51.5 ± 9.6 | 55.2 ± 10.0 | 50.6 ± 9.7 | 53.3 ± 10.2 | 280 (140 pairs) from 6 cells |
| Random | Spc25-EGFP | 9G3 | N.A. | 50.8 ± 6.7 | 55.3 ± 7.7 | 50.3 ± 6.7 | 54.2 ± 7.8 | 120 from single cell |

DOI: https://doi.org/10.7554/eLife.32418.013

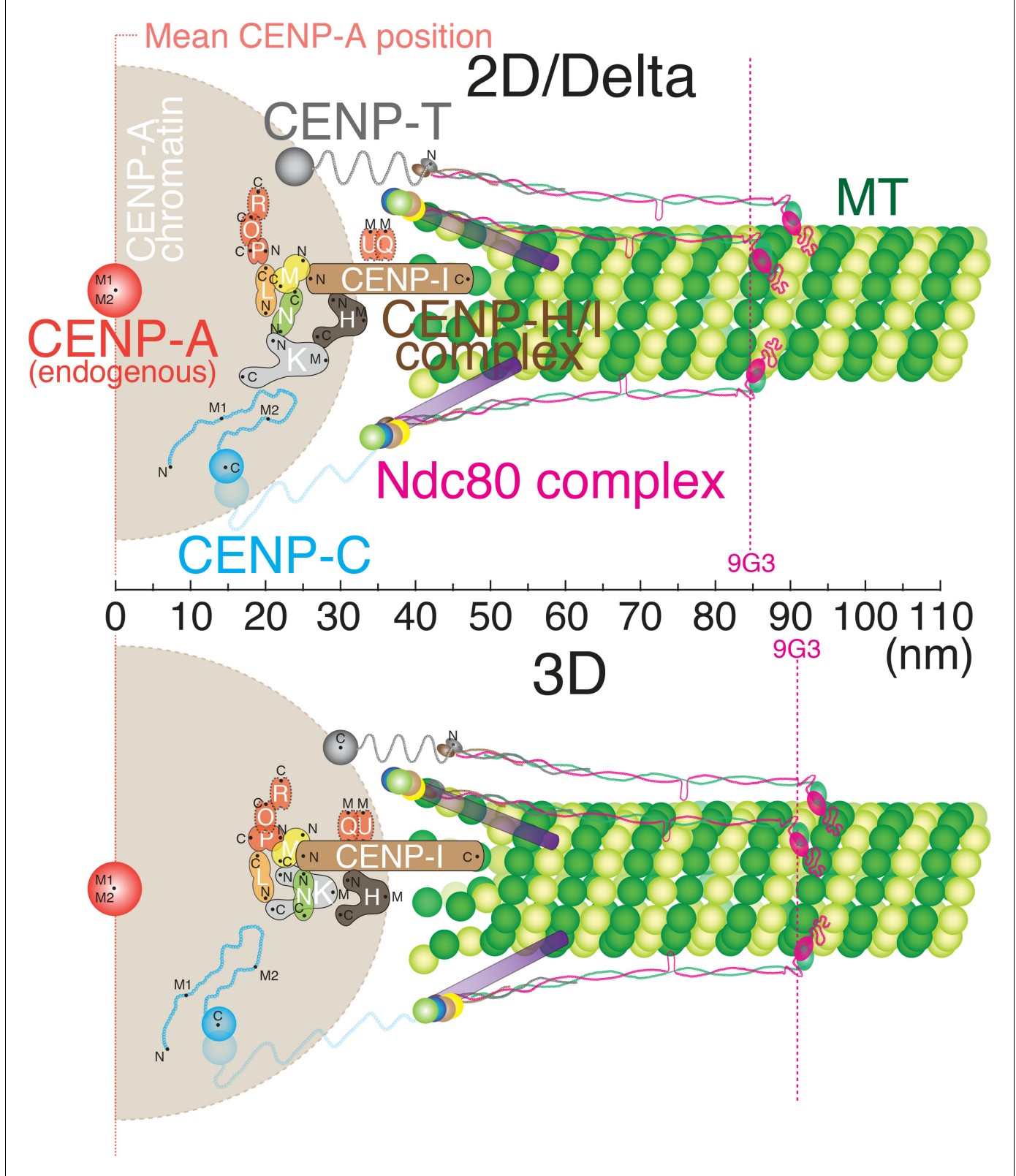

**Figure 5.** Schematic core-kinetochore drawing measured by 2D/Delta or 3D co-localization method. Schematic picture of the mean positions of protein labels within the core kinetochore protein architecture measured using ML2D/Delta (top) and ML3D (bottom) methods. N or C are the mean positions of N-terminal or C-terminal EGFP fusions or specific antibodies. M indicates the mean position measured using polyclonal antibodies whose epitope(s) are not precisely known.

*Figure 5 continued on next page*

*Figure 5 continued*

DOI: https://doi.org/10.7554/eLife.32418.014

The following source data is available for figure 5:

**Source data 1.** List for core-kinetochore protein measurements.

DOI: https://doi.org/10.7554/eLife.32418.015

**Source data 2.** List of primary antibodies.

DOI: https://doi.org/10.7554/eLife.32418.016

mean separation between green and red labels. This comparison also indicated that there was something about specimen preparation causing variation in z-axis CA.

To explore possible optical factors, we used a mathematical analysis reported previously (*Gibson and Lanni, 1992*). This analysis calculates, relative to the design values for the objective, changes in optical path for light coming from a fluorescent point as a function of the refractive index and thickness of the specimen media, the coverslip thickness, and immersion oil, as well as wavelength. It showed that the amount of z-axis CA is highly sensitive to changes in the thickness of the coverslip. We tested this possibility by using different thickness of coverslips. We did most of our experiments reported in this paper using # 1.5 coverslips. We measured their thickness with a micrometer, and the mean value was 0.17 mm with a range of 0.16 to 0.19 (*Figure 6*, n > 100). For #1 coverslips, the mean was 0.15 mm and for #2 coverslips it was 0.22 mm (*Figure 6*). The mean Sz (Z-offset) for CENP-T(M) to Hec1-9G3 showed ~±10 nm for #1.5 coverslips, −29 to −19 nm for #1 coverslips, and 29 to 46 nm for # 2 (*Table 3B*). The plot in *Figure 6—figure supplement 1* illustrates the sensitivity of Z-offset to coverslip thickness. A steep slope is seen when measured Z-offset values in *Table 3B* are plotted as a function of the mean thickness we measured for #1, #1.5 and #2 coverslips. Based on these experimental results, we suspect that unusually large Z-offset values (>25 nm) measured in our experiments (*Figure 5—source data 1*) were largely produced by coverslip thicknesses much thinner or thicker than 0.17 mm.

In our study, we were careful to include similar numbers of kinetochores above and below the spindle equator. This is because kinetochores and their kMTs become tilted progressively with their radial distance away from the spindle axis (*Figure 6—figure supplement 2A*) (*Smith et al., 2016; Wan et al., 2009*). Both the inner and outer domains of metaphase kinetochores typically have their faces perpendicular to their kMTs (*Wan et al., 2009*). If kinetochores are only measured on the bottom half of the spindle, the mean value of Sz will not be zero because the upward kinetochore tilt in the lower half spindle is not canceled by the downward tilt from kinetochores in the upper half-spindle (*Figure 6—figure supplement 2A–B*). This kinetochore tilt had no effect on measured 3D mean separation since values from only kinetochores in the upper half of the spindle yielded similar results

**Table 3.** (A) List of Delta, S(2D), S(3D), Z-offset, and zS(3D) (Z-offset subtracted) for the mean separation between CENP-T(M) and Hec1(9G3) in samples with or without large Z-offset.

(B) List of Delta, S(2D), S(3D), Z-offset, and zS(3D) in samples with #1, #1.5, and #2 coverslips. The value zS(3D) was calculated from the S(3D) data after subtraction of the mean Z-offset in the middle of the spindle from the Sz values. All values were mean ±SD.

| A | | Green | Red | Delta nm) | S(2D)(nm) | S(3D) (nm) | Z-offset (nm) | zS(3D) (nm) | |
|---|---|---|---|---|---|---|---|---|---|
| | Example 1 | CENP-T (M) | 9 G3 | 59.5 ± 6.3 | 60.8 ± 8.8 | 102.4 ± 26.7 | 79.7 ± 32.9 | 68.9 ± 10.2 | |
| | Example 2 | CENP-T (M) | 9 G3 | 57.3 ± 7.3 | 60.1 ± 11.7 | 64.8 ± 11.3 | 6.9 ± 23.2 | 64.5 ± 10.8 | |
| B | | Thickness | Green | Red | Delta nm) | S(2D)(nm) | S(3D) (nm) | Z-offset (nm) | zS(3D) (nm) |
| | Experiment 1 | #1 | CENP-T (M) | 9 G3 | 60.4 ± 6.9 | 63.5 ± 12.4 | 75.2 ± 17.6 | −28.6 ± 31.1 | 70.4 ± 14.2 |
| | Experiment 2 | #1 | CENP-T (M) | 9 G3 | 59.2 ± 10.0 | 62.6 ± 11.7 | 72.6 ± 16.3 | −18.3 ± 33.9 | 70.6 ± 14.6 |
| | Experiment 1 | #1.5 | CENP-T (M) | 9 G3 | 57.3 ± 7.3 | 60.1 ± 11.6 | 64.8. ± 11.3 | 6.9 ± 23.2 | 64.5 ± 10.8 |
| | Experiment 2 | #1.5 | CENP-T (M) | 9 G3 | 57.0 ± 7.1 | 60.6 ± 11.1 | 64.0 ± 11.6 | 6.0 ± 20.0 | 63.8 ± 11.2 |
| | Experiment 1 | #2 | CENP-T (M) | 9 G3 | 59.6 ± 7.7 | 62.1 ± 11.9 | 82.1 ± 27.2 | 46.4 ± 36.4 | 69.6 ± 21.9 |
| | Experiment 2 | #2 | CENP-T (M) | 9 G3 | 59.8 ± 7.4 | 62.5 ± 10.9 | 74.2 ± 16.2 | 28.7 ± 30.2 | 69.4 ± 10.8 |

DOI: https://doi.org/10.7554/eLife.32418.017

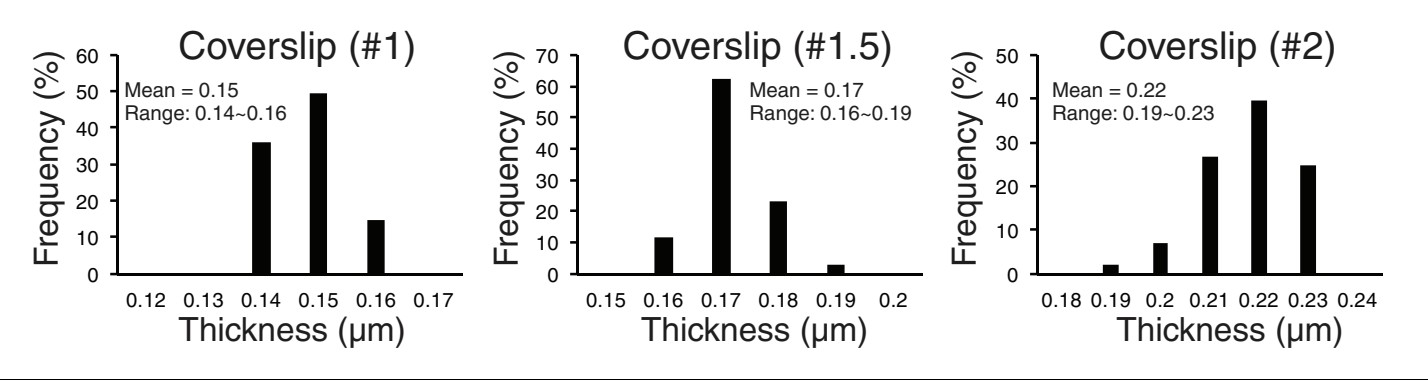

**Figure 6.** Accuracy of separation measurements depends critically on mean z-axis CA (Z-offset) produced by coverslip thickness variations from standard value. Measured variability of coverslip thickness within #1, #1.5, and #2 coverslips.

DOI: https://doi.org/10.7554/eLife.32418.018

The following figure supplements are available for figure 6:

**Figure supplement 1.** Z-offset changes dependent on coverslip thickness.
DOI: https://doi.org/10.7554/eLife.32418.019

**Figure supplement 2.** Z-axis offset from kinetochore tilt.
DOI: https://doi.org/10.7554/eLife.32418.020

as from kinetochores in the lower half of the spindle (data not shown). Unlike 3D separation, the mean tilt angle β (but not angle α) was sensitive to unequal numbers of kinetochores above and below the spindle equator. The mean angle β is near zero in the middle of the spindle and reaches mean values of about 20–25 degrees at the bottom and top of the spindle (*Figure 6—figure supplement 2C–D*).

## PFA fixation does not cause shrinkage, but pre-extraction or detergent in PFA solution causes shrinkage of cellular structure

A recent study raised the issue that PFA fixation induced shrinkages compared to GA fixation (*Magidson et al., 2016*). GA has two aldehyde radicals and provides more robust fixation compared to PFA fixation. Because of this reason, GA fixation is usually used for electron microscopy. On the other hand, GA fixation exhibits strong auto fluorescence, and induces significantly less reactivity between cellular antigen and antibodies compared to PFA fixation. For immunofluorescence, a non-ionic detergent, such as Triton-X 100 or NP40, is often used to improve antibody penetration into the cell before, with, or after fixation. It is uncertain how these pre-permeabilization processes affect cellular architecture. For example, *Magidson et al. (2016)* concluded PFA fixation causes shrinkage. However, they actually supplemented 1% Triton-X 100 in their PFA fixation solution. Our standard method was primary fixation by PFA without detergent (Method one in *Table 4*), a method significantly different than the PFA fixation method used by (*Magidson et al., 2016*).

To test the effect of fixation and detergent for antibody reaction and separation measurements, we tested nine different fixation methods (*Table 4*, and Materials and methods). We used CENP-T (M) and Hec1-9G3 antibodies for this comparison (*Figure 7A*). These antibodies were the exact same antibodies used in *Figure 3* in this study, and in previous reports (*Fuller and Straight, 2012*; *Magidson et al., 2016*; *Suzuki et al., 2014*; *Wan et al., 2009*). We first compared CENP-T(M) and Hec1-9G3 signal levels at kinetochores in the different methods (*Figure 7B*). Interestingly, both CENP-T(M) and Hec1-9G3 signal levels in GA fixation samples (Methods 4–7) were significantly reduced by ~35% to 60% compared to PFA fixation (Method 1), PFA with detergent fixation (by 20–30%), or MeOH fixation (~20%) (*Figure 7B*). However, local background levels (surrounding area of kinetochore) were not significantly different between all fixation methods, indicating that GA fixation significantly reduced the signal to noise ratio (S/N) compared to PFA or MeOH fixation (*Figure 7B*).

Next, we measured mean separation between CENP-T(M) and Hec1-9G3. Surprisingly, we could not find any significant differences between PFA fixation (Methods 1–2) and GA fixation (Methods 4–7). However, we observed significant shrinkage when samples were fixed by Methods 3, 8, and 9

**Table 4.** Summary of fixation methods used in *Figure 7*.
Detailed protocols for the different fixation methods used in *Figure 7*. Method 1 is the protocol used in this study.

| Method | Pre-fixation | Pre-permeabilization | Fixation | Quenching | Post-permeabilization | Primary antibody | Secondery anitbody | References (use for nm-resolution analysis) |
|---|---|---|---|---|---|---|---|---|
| Method 1 | | | 3% PFA in PHEM (37°C) | | 0.5% NP40 | 37°C for 90 min | 37°C for 90 min | *Suzuki et al., 2011* ;*Suzuki et al., 2014* ;*Tauchman et al., 2015*; this study |
| Method 2 | 3% PFA in PHEM (37°C) | 0.5% triton in PHEM (37°C) | 3% PFA in PHEM (37°C) | | 0.5% NP40 | 37°C for 90 min | 37°C for 90 min | *Wan et al. (2009)*; *Varma et al., 2013* |
| Method 3 | | 1% triton in PHEM (37°C) | 3% PFA in PHEM (37°C) | | 0.5% NP40 | 37°C for 90 min | 37°C for 90 min | |
| Method 4 | | | 1% GA in PHEM (37°C) | NaBH$_4$ | 0.5% NP40 | 37°C for 90 min | 37°C for 90 min | |
| Method 5 | | | 1%GA w 1% triton in PHEM (37°C) | NaBH$_4$ | 0.5% NP40 | 37°C for 90 min | 37°C for 90 min | *Magidson et al. (2016)* |
| Method 6 | | 1% triton in PHEM (37°C) | 1% GA in PHEM (37°C) | NaBH$_4$ | 0.5% NP40 | 37°C for 90 min | 37°C for 90 min | |
| Method 7 | | 1% triton in PHEM (37°C) | 1% GA w 3% PFA in PHEM (37°C) | NaBH$_4$ | 0.5% NP40 | 37°C for 90 min | 37°C for 90 min | |
| Method 8 | | | MeOH (−20°C) | | 0.5% NP40 | 37°C for 90 min | 37°C for 90 min | |
| Method 9 | | | 3%PFA w 1% triton in PHEM (37°C) | | 0.5% NP40 | 37°C for 90 min | 37°C for 90 min | *Magidson et al. (2016)* |

DOI: https://doi.org/10.7554/eLife.32418.021

(*Table 5*). Interestingly, we could not see any shrinkage when we fixed by GA with Triton X-100, Triton X-100 treatment before GA or GA/PFA (Methods 5–7). Although there are no differences between PFA and GA fixation for the separation between CENP-T(M) and Hec1-9G3, we found that pre-extraction by Triton X-100 or PFA fixation with Triton X-100 induced significant shrinkage compared to GA with Triton X-100. This result is consistent with the report of *Magidson et al. (2016)*. However, our standard PFA fixation method (Method 1) did not produce any shrinkage.

## Computer simulations for understanding the method and its limitations

The method using mean values of CA to correct chromatic aberration is widely used, but the limitations of this method are not well established. We evaluated the method by computer simulations. Critical factors are SDs of centroid determination (CDsd) and CA (CAsd). In our simulations, kinetochores were located at 10 pixel intervals in both the X and Y axis directions for the typical image region of 0–280 pixels in the X- direction and 0 to 190 pixels in the Y-direction (*Figure 8*). At each kinetochore position (p), the separation vector extended from the origin to a length S in the x direction. The green (Xgp, Ygp, Zgp) values at the origin had a mean of zero plus a CDsd determined by sampling CDsd times a normalized random number for the x, y, and z directions (*Figure 8*). The red (Xrp, Yrp, Zrp) values included S in the x-direction and Z-offset in the Z direction. They also included a value for the CA at each kinetochore position, CAp, plus CDsd for each direction (*Figure 8*). Two methods were used to generate the CAp values. In the first, for each x, y, and z direction, CAp was calculated from the measured mean CA plus the CAsd times a normalized random number. In the second, CAp for the x and y directions were calculated from least square fits to the CA cellular data because of the 30 nm and 20 nm decrease respectively across the measurement field (*Figure 2—figure supplement 3*). However, this second procedure gave identical results as the first method outlined in *Figure 8*, which is the CA correction method we used experimentally. Mean and SD values for S(2D), S(3D), ML2D, ML3D, angle α, and angle β were calculated using the same equations as we did in analysis of cellular data (*Figures 1E* and *3C*).

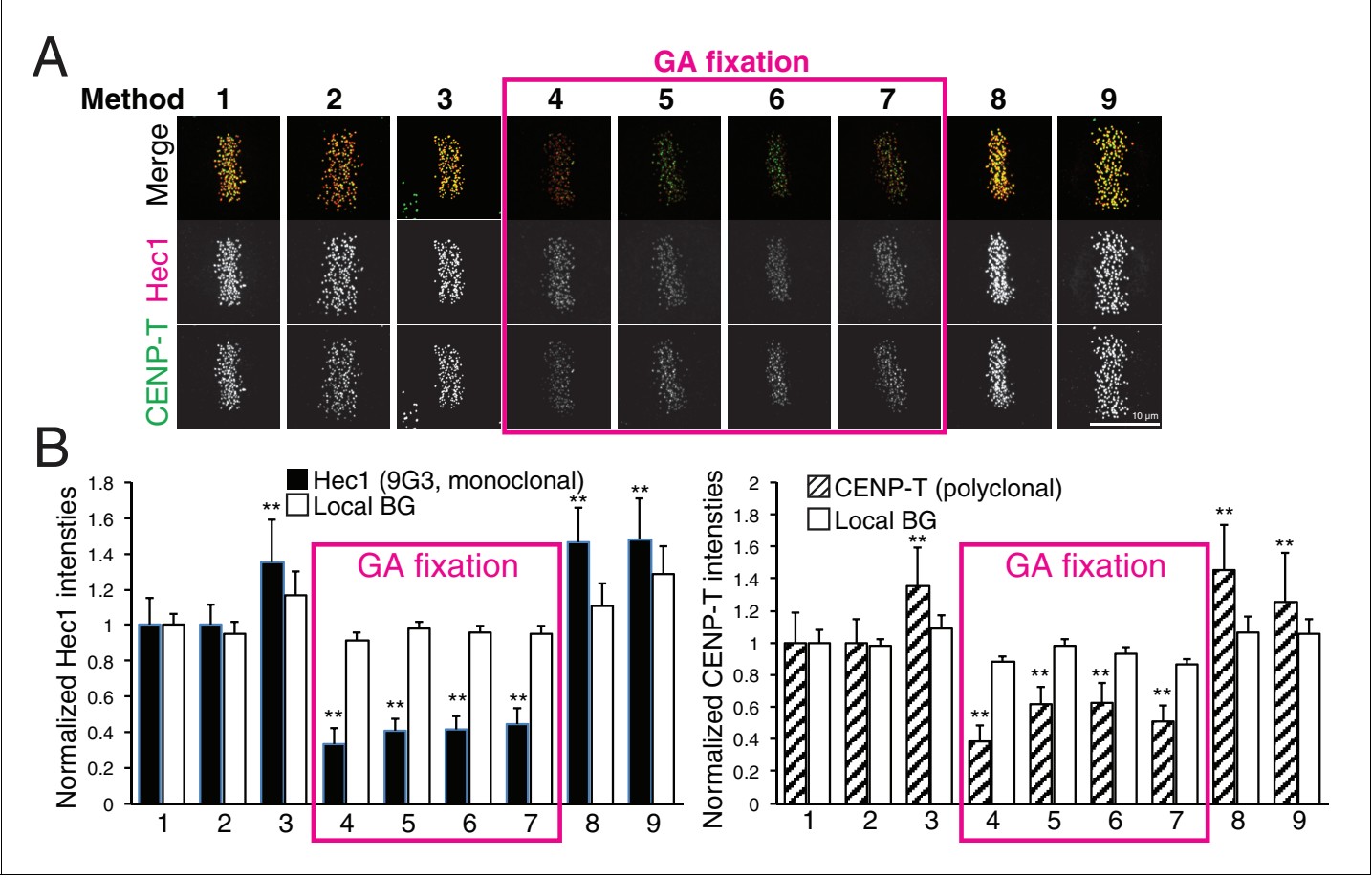

**Figure 7.** Comparison of various fixation methods for signal intensity and mean fluorescence co-localization analysis. (**A**) Representative immunofluorescent images stained by CENP-T(M) and Hec1-9G3 antibodies using the fixation protocols listed in *Table 4*. (**B**) The plots of normalized signal intensity of Hec1-9G3 (left) and CENP-T (right) at kinetochores using the fixation protocols listed in *Table 4*. The signal intensities were normalized for the mean values of PFA fixation (~20,000 integrated fluorescence counts above local background for Method 1). All values were mean ±SD. n > 200 individual kinetochores for each experiment. **p<0.05 (t-test).

DOI: https://doi.org/10.7554/eLife.32418.022

To obtain an estimate of the error contributed from the CDsd, we used results from Delta analysis which in principle have local CA correction. For example, the Delta value measured for CENP-T(M) to Hec1-9G3 is 60 ± 7.5 nm. In the simulation, we made the CAsd = zero, and asked how much CDsd is needed to make the measured Delta SD =±7.5 for S = 60 nm. The result was CDsd (x, y, z) = (±4,±4,±8) (*Table 6A*). Because resolution in the z direction is half that in the x and y directions, we set CDsd of z = 2 * CDsd of x or y. Simulations that include the CDsd, but exclude CAsd, yielded mean values for ML3D and ML3D nearly identical to the input 60 nm value (*Table 6A*). This result indicates that the SD in our centroid measurement is very small and only slightly enhances the 3D measurements from the true value.

The measured CAsd (x, y, z) = (±9.1, ±7.5, ±17.6 nm) are only about twice those for centroid determination, and make only a slightly larger contribution to errors in measured S(2D), S(3D), ML2D, ML3D, angle α, and angle β (*Table 6A*). For S = 60 nm, the mean of S(2D) and S(3D) are increased by a CDsd and CAsd to 61 and 65 nm, but this 1–5 nm error is mostly corrected by application of maximum likelihood fit to yield ML2D = 60 ± 11 nm and ML3D = 62.3 ± 12 nm (*Table 6A*). This result shows that our method of subtracting the mean values of CA combined with maximum likelihood data analysis can yield accurate mean 3D values for separation vector length.

To see how the CDsd and CAsd of our measurement method affect the SDs in the angles α and β, we entered into the simulation constant values for S between zero and 100 nm (*Table 6B*). The SD

**Table 5.** The list of measured values for Delta, ML2D, ML3D, dDelta, dML2D, and dML3D using protocols in *Table 4*.
The dDelta, dML2D, dML3D were differences from Method 1 (PFA fixation). All values were mean ±SD. n > 200 individual kinetochores
for each experiment.

| | Red-Green = mean ± S.D. nm | | | | | |
| --- | --- | --- | --- | --- | --- | --- |
| | Delta (nm) | ML2D (nm) | ML3D (nm) | dDelta (nm) | dML2D (nm) | dML3D (nm) |
| Method 1 | 57.2 ± 7.3 | 59.1 ± 11.6 | 62.6 ± 11.5 | 0 | 0 | 0.0 |
| Method 2 | 56.8 ± 6.0 | 58.6 ± 8.8 | 61.3 ± 8.6 | −0.4 | −0.5 | −1.3 |
| Method 3 | 50.1 ± 7.1 | 52.8 ± 8.5 | 56.4 ± 8.2 | −7.1 | −6.3 | −6.2 |
| Method 4 | 58.7 ± 9.1 | 61.5 ± 12.4 | 63.1 ± 22.9 | 1.5 | 2.4 | 0.5 |
| Method 5 | 55.2 ± 8.9 | 56.9 ± 11.7 | 61.3 ± 15.5 | -2 | −2.2 | −1.3 |
| Method 6 | 57.2 ± 9.0 | 60.0 ± 12.9 | 63.5 ± 21.6 | 0 | 0.9 | 0.9 |
| Method 7 | 54.0 ± 10.0 | 57.2 ± 12.1 | 57.7 ± 12.2 | −3.2 | −1.9 | −4.9 |
| Method 8 | 46.4 ± 5.3 | 48.3 ± 7.7 | 52.0 ± 7.7 | −10.8 | −10.8 | −10.6 |
| Method 9 | 51.8 ± 7.4 | 53.8 ± 9.7 | 56.7 ± 10.0 | −5.4 | −5.3 | −5.9 |

DOI: https://doi.org/10.7554/eLife.32418.023

in the angles $\alpha$ and $\beta$ are largest for small separations (*Table 6B*). For example, they were 9 and 19 degrees (S = 60 nm) or 6 and 11 degrees (S = 100 nm). These results indicate that our CDsd and CAsd caused about half to two-thirds of the SD measured in cells for separations of 60 nm (SDs of angles $\alpha$ and $\beta$ were 21 and 23 degrees, respectively; CENP-T-Hec1-9G3 in *Figure 3C*) or 90 nm

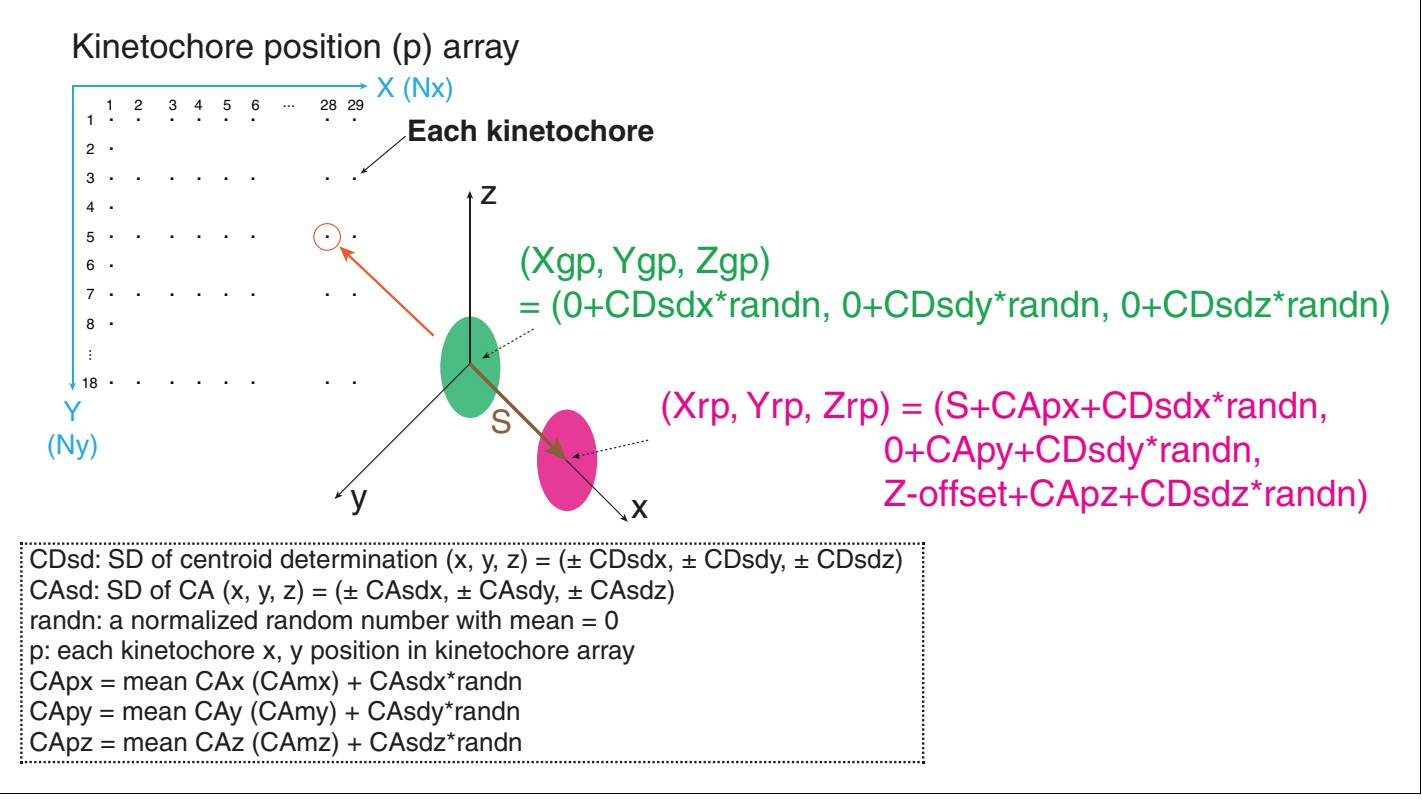

**Figure 8.** Diagram of simulation model. We used a 290 × 180 pixel region (x and y) and kinetochores at positions (p) separated by 10 pixels in each direction. All simulated values S(2D), S(3D), angle $\alpha$, and angle $\beta$ were averaged from 29 × 18 = 551 kinetochores. The maximum likelihood fit was applied to the raw simulated S(2D) and S(3D) values in the same way as for the experimental analysis in this study.
DOI: https://doi.org/10.7554/eLife.32418.024

**Table 6.** Computer simulations to determine measurement SD and to test the accuracy of using mean CA for CA correction. (A) Comparison of ~60 nm experimental results to simulated results for a fixed S value of 60 nm, the estimated CDsd for our measurements, and with or without CAsd. (B) Simulated S(3D), and ML3D values for different input S values with fixed angles (0°), CDsd (x, y, z) = (±4, ±4, ±8) and CAsd (x, y, z) = (±9.1, ±7.5, ±17.6). All simulated values were mean ±SD.

| A | | Delta (nm) | S(2D) (nm) | S(3D) (nm) | ML2D (nm) | ML3D (nm) | α (°) | β (°) |
|---|---|---|---|---|---|---|---|---|
| Experiment | CENP-T(M)-Hec1-9G3 | 60 ± 7.5 | 63.1 ± 11 | 68.6 ± 12.3 | 62.1 ± 11.3 | 66.3 ± 12.3 | −0.3 ± 21 | −1.8 ± 23 |
| Simulation (S = 60 nm) | SDc (x, y,z) = (±4, ±4, ±8) | | | | | | | |
| | No CAsd | 60.4 ± 5.7 | | 61.5 ± 5.8 | 60.4 ± 5.7 | 61.5 ± 5.8 | 0.1 ± 5.4 | 0 ± 10.7 |
| | CAsd (x, y, z) = (±9.1, ±7.5, ±17.6) | N/A | 60.7 ± 10.9 | 64.5 ± 11.3 | 59.7 ± 11 | 62.3 ± 11.5 | −0.2 ± 9.4 | −0.1 ± 19.4 |
| B  input values | | | simulation results | | | | | |
| S ± 0 nm | α (±0°) | β (±0°) | α° | β° | S(3D) (nm) | ML3D (nm) | | |
| 0 | 0 | 0 | 0.3 ± 51.2 | 1.8 ± 51.8 | 22 ± 10.5 | 0 ± 14.1 | | |
| 10 | 0 | 0 | −2.4 ± 44.8 | −3.2 ± 48.8 | 24.6 ± 10.9 | 0 ± 15.5 | | |
| 15 | 0 | 0 | 1.1 ± 36 | −0.6 ± 43.5 | 26.5 ± 11.1 | 0 ± 16.6 | | |
| 20 | 0 | 0 | 0.9 ± 30.9 | −2.4 ± 40.3 | 29.5 ± 11.4 | 20.7 ± 13.8 | | |
| 40 | 0 | 0 | 0.8 ± 14.8 | −0.3 ± 27.2 | 47 ± 12.3 | 43.1 ± 12.9 | | |
| 60 | 0 | 0 | 0.1 ± 9.1 | −0.5 ± 19.3 | 63.9 ± 11.4 | 61.8 ± 11.5 | | |
| 80 | 0 | 0 | −0.1 ± 6.6 | 1 ± 14.3 | 83.1 ± 11.2 | 81.6 ± 11.3 | | |
| 100 | 0 | 0 | 0.2 ± 5.5 | 0.3 ± 11.4 | 102.9 ± 11.1 | 101.7 ± 11.1 | | |

DOI: https://doi.org/10.7554/eLife.32418.025

(SDs of angles α and β were 16 and 22 degrees, respectively; CENP-A-Hec1-9G3 in *Figure 6—figure supplement 2C–D*). These simulations also showed that our SDs inhibit maximum likelihood fit below S = ~18 nm (*Table 6B*). These simulation results indicate that much of our measured SD in the cell is not caused by variability between different kinetochores, but by the SDs in centroid measurement and CA correction.

## Discussion

There are several key factors that influence the accuracy of the 3D fluorescence co-localization method we used for determining mean separation distances (all factors we identified are listed in *Figure 1—source data 1*). High S/N and bright specimens are needed for small CDsd. The measurement field should be restricted in area to keep the CAsd small. The correct coverslip thickness should be used to prevent Z-offset. These procedures are important because computer simulations show that mean 2D/3D separations and SDs in angles (α and β) all increase from their true values as a power function of the magnitudes of CDsd, CAsd, and Z-offset (*Figure 9A–C*). In addition, the use of the maximum likelihood method corrects the overestimate of mean separations obtained from S (2D) and S(3D) (*Churchman et al., 2005*).

Z-offset is a major source of error only if it is not produced by the specimen. As a general rule, we do not recommend subtracting Z-offset from Sz data since there could be a specimen contribution. When using our method, we recommend repeating the specimen measurement with a 0.17 mm coverslip if Z-offset is larger than ±25 nm to make sure there is not an unknown specimen contribution. Computer simulations of the measurement method showed that only absolute values of Z-offset >30 nm make 3D mean separation measurements significantly larger (*Figure 9C*). Also, our method produces nearly identical results from multiple samples. *Figure 5—source data 1* has measurements between CENP-A and Hec1-9G3 from four different coverslip preparations. Their Z-offsets varied between −17 and 4.4 nm, while their ML3D mean values varied between 86 and 93 nm.

*Smith et al., 2016* reported that the mean 3D separation between endogenous CENP-A and the C-terminus of Ndc80 was 81.7 ± 0.7 nm SEM (n = 3302, SD =±40 nm), about 42 nm larger than our measured value of 40 nm ± ~ 10 nm (SD) (*Table 7*). *Table 7* also provides three more examples showing differences of 24 nm to 60 nm between their and our mean 3D separation measurements.

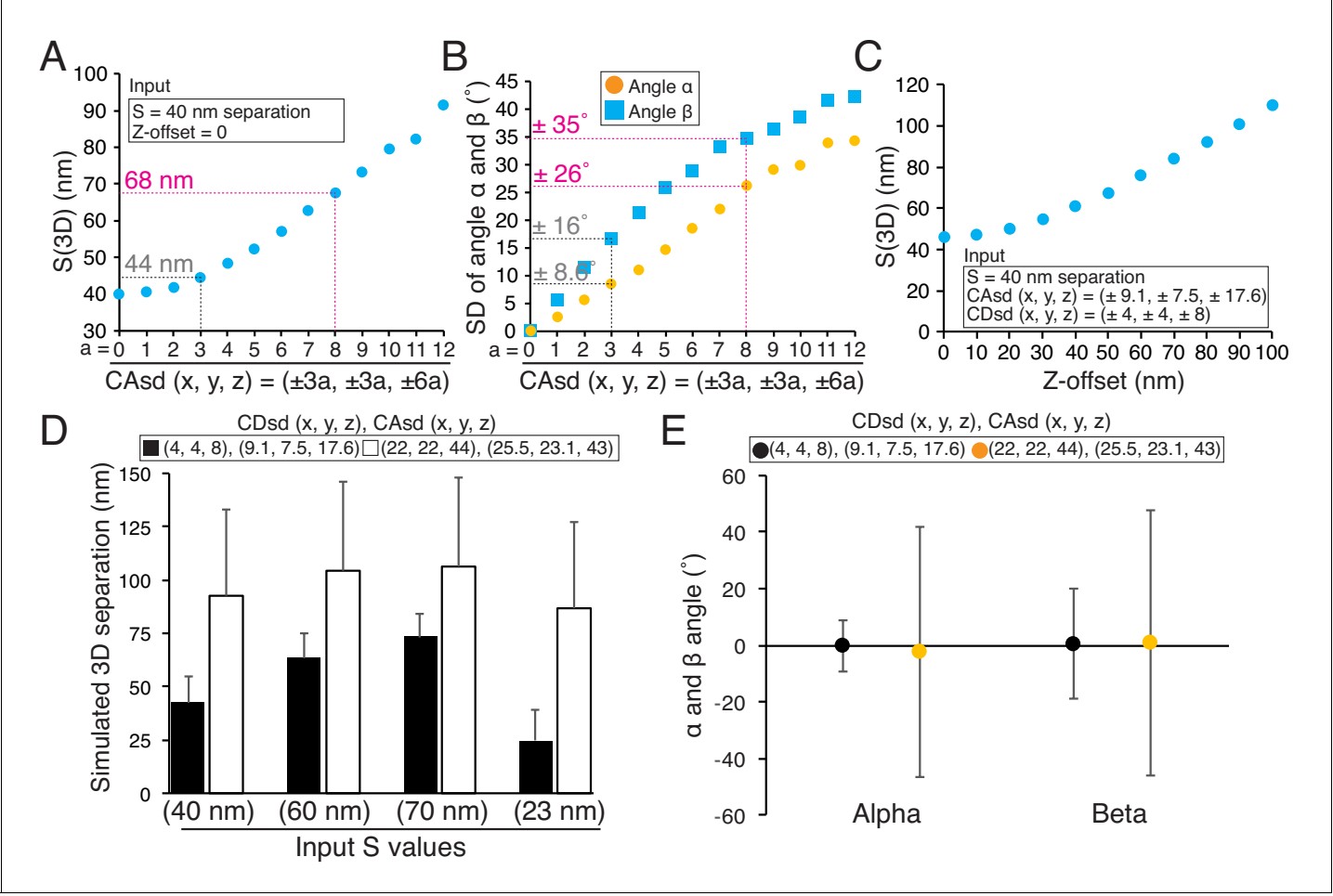

**Figure 9.** Origins of error in mean separation measurements and comparison of our results to previous reports. (**A**) Plots of simulated S(3D) for 40 nm separation with increasing values of CAsd (SD of CA) where the mean CA values were held constant at our measured values of 13.1, 15.8, and 3.5 nm respectively. Grey dashed line is nearly identical to the CAsd for our measurements and the red dashed line is nearly identical to the CAsd for the *Smith et al., 2016* measurements. (**B**) Plots of SD of simulated angle α and β for 60 nm separation using the same condition in (**A**). (**C**) Plots of simulated S(3D) for S = 40 nm separation for increasing values of Z-offset and the CAsd reported by *Smith et al. (2016)*. (**D**) The simulated 3D separation using our SDs (black bars) or in *Smith et al. (2016)* (white bars). The true separations (S values) are 40, 60, 70, and 23 nm from left to right, which correspond to the mean values from Delta analysis in *Table 7*. (**E**) The simulated mean angle α and β with ±SD for the conditions in (**E**). Note, the input mean angle α and β were zero, and S = 60 nm.

DOI: https://doi.org/10.7554/eLife.32418.026

The following source data and figure supplement are available for figure 9:

**Source data 1.** Estimation of SD for centroid determination (CDsd).
DOI: https://doi.org/10.7554/eLife.32418.028
**Source data 2.** Estimation of kinetochore-kinetochore variability (Ksd) in our measurements.
DOI: https://doi.org/10.7554/eLife.32418.029
**Figure supplement 1.** About 10 ~15 kinetochore measurements are sufficient to obtain a mean value close to the mean separation values ± SD from>200 kinetochores.
DOI: https://doi.org/10.7554/eLife.32418.027

Since *Smith et al., 2016* used mean CA values for CA correction as we did in this study, we investigated the possibility that significantly higher CDsd and CAsd compared to our values might be responsible for their larger 3D separation values. We estimated the CDsd from the SD of their Delta measurements for CENP-A to Spc24, which was ~35 nm, compared to our value of 6 nm (*Figure 9— source data 1*) (*Smith et al., 2016*; *Suzuki et al., 2014*). This analysis yielded values for CDsd about five times larger than ours (22 nm, 22 nm, and 44 nm compared to 4 nm, 4 nm and 8 nm in the x, y

**Table 7.** List of four examples of Delta and 3D mean separation measurements with ±SD from this study and *Smith et al., 2016*.

| | | | | Delta (Mean ± SD nm) | 3D separation (Mean ± SD nm) |
|---|---|---|---|---|---|
| 1 | *Smith et al., 2016* | EGFP-CENP-A | Ndc80(C-term) | 58 ± 35 (n = 1002,±1.1 (SE)) | 98 ± 58 (n = 4291,±0.9 (SE)) |
| | *Smith et al., 2016* | endogeous CENP-A | Ndc80(C-term) | N/A | 84 ± 40 (n = 3302,±0.7 (SE)) |
| | This study | endogeous CENP-A | CENP-T (N-term) | 40 ± 6 (n = 110) | 46 ± 12 (n = 220) |
| 2 | *Smith et al., 2016* | endogenous CENP-C | Ndc80 (N-term) | N/A | 103 ± 74 (n = 7476,±0.8 (SE)) |
| | This study | endogenous CENP-C | Hec1-9G3 | 58 ± 5 (n = 62) | 67 ± 8 (n = 124) |
| 3 | *Smith et al., 2016* | EGFP-CENP-C | Ndc80 (N-term) | N/A | 101 ± 76 (n = 1769,±1.8 (SE)) |
| | This study | EGFP-CENP-C | Hec1-9G3 | 69 ± 9 (n = 102) | 77 ± 15 (n = 124) |
| 4 | *Smith et al., 2016* | EGFP-CENP-O | Ndc80(C-term) | N/A | 81 ± 52 (n = 1881,±1.2 (SE)) |
| | This study | EGFP-CENP-O | Ndc80(C-term) | 23 ± 11 (n = 97)* | 21 ± 16 (n = 194)* |

*These values were calculated by (EGFP-CENP-O - Hec1-9G3) minus (Ndc80-EGFP - Hec1-9G3).

DOI: https://doi.org/10.7554/eLife.32418.030

and z directions) (*Figure 9—source data 1*). Such high CDsd values could explain why their Delta measurements are significantly larger than ours (*Figure 9—source data 1*). Their reported CAsd was also significantly larger than ours (26 nm, 23 nm and 43 nm compared to 9 nm, 7.5 nm and 18 nm in the x, y and z directions) (*Smith et al., 2016*). Results of simulations in *Figure 9D* using the values we estimated for their CDsd and their reported CAsd produce results nearly identical to their mean separation measurements, and show how the differences in the SDs between their and our measurements could be responsible for their much larger values. In addition, the higher levels for CDsd and CAsd are likely responsible for the very large SDs they measured for the angles α and β compared to ours (*Figure 9E*). In our measurements, large values of Z-offset were rare (*Figure 5—source data 1*); we have no information to judge how much other sources of error like Z-offset, contributed to the *Smith et al. (2016)* measurements.

Human kinetochores are built from multiple copies of core-kinetochore proteins (*Suzuki et al., 2015*). A previous study has shown that the accuracy of separation measurements is sensitive to the spatial staggering of the labeled molecules along the kinetochore inner to outer axis only for stagger greater than 200 nm, which has the possibility of producing two fluorescent peaks instead of a single peak if staggering is not random (*Joglekar et al., 2009*). All kinetochores labeled by specific antibodies or EGFP in our study exhibited a Gaussian distribution with a single peak. This indicates that our centroid measurements correspond to the mean position of the fluorescently labeled protein epitopes within human kinetochores.

*Magidson et al. (2016)* questioned whether the mean separation measurements averaged from multiple kinetochores (typically 200–400 kinetochores from 5 to 20 cells in our study) hide variation in the position of proteins relative to each other between different kinetochores (*Magidson et al., 2016*). We tried to address this question in several ways. First, we found that mean measurements of individual kinetochores (not sister pairs) within a single cell yielded the identical value at metaphase as the mean obtained from kinetochores (sister pairs) within six different cells (*Table 2*). Thus, variations between cells or selected kinetochore measurements like in the Delta analysis is not an issue. Second, as expected for the Gaussian distribution measured for separation values (e.g. *Figure 3B*), we found that only 10–15 kinetochores were needed to be averaged to achieve the mean and SD typical of >150 kinetochores (*Figure 9—figure supplement 1*). Third, the SD of our 2D/3D measurements are small, typically less than 12 nm, and a significant part of this SD must be from the CDsd and CAsd (*Table 6A* and see above discussion) and not from differences between kinetochores. To see this more clearly, we used the same simulations as in *Figures 8–9* but with mean kinetochore-to-kinetochore variability (Ksd = ±0, ±5, ±10, or ±15 nm) (*Figure 9—source data 2*). Simulation results showed that Ksd of about ±5 nm or less plus measurement CDsd and CAsd yielded the experimental mean SD value of 12.5 nm (*Figure 9—source data 2*). These results imply that the vast majority of kinetochores at metaphase have a similar protein architecture.

We found that our PFA fixation does not shrink kinetochore structure and produces exactly the same Delta, 2D, and 3D mean separation values obtained by the GA fixation. A major problem with GA fixation is that it interferes with antibody labeling, preventing immunofluorescence with many

antibodies, and reduced fluorescence for others like those used in *Figure 7*. In addition, our data indicate that primary GA fixation may not be suitable for analysis of molecular distribution by immuno-electron microscopy because of poor recognition by antibodies. For immuno-electron microscopy, GA fixation should be used as the post-fixation solution, but only after the antibody reaction has been performed.

This study provides an optimized 3D fluorescence co-localization method with error analysis. It can achieve ~10 nm accuracy for mean separations larger than 18 nm. Our mean 3D fluorescence co-localization measurements for human kinetochore protein architecture are close to the mean values obtained using local correction of CA (*Wan et al., 2009*; *Fuller and Straight, 2012*) and successfully predict the known structural lengths of the Ndc80 complex components (*Ciferri et al., 2008*; *Valverde et al., 2016*; *Musacchio and Desai, 2017*). This optimized 3D co-localization method is a useful tool for broad research fields that need to determine mean protein architecture with nm-accuracy.

## Materials and methods

### Cell culture

HeLa cells were cultured in Dulbecco's modified Eagle's medium (Gibco) supplemented with 10% fetal bovine serum (Sigma), 100 U/ml penicillin and 100 mg/ml streptomycin at 37 °C in a humidified atmosphere with 5% $CO_2$ (*Suzuki et al., 2015*). siRNA resistance EGFP-Ndc80, Ndc80-EGFP, EGFP-Nuf2, Nuf2-EGFP, EGFP-Spc24, Spc24-EGFP, EGFP-Spc25, and Spc25-EGFP constructs were linearized by restriction enzymes, and then purified plasmids were transfected using Effectene regents (Qiagen) according to the manufacturer's instruction. The transfected cells were selected by Geneticin (Invitrogen) 48 hr after transfection. We collected more than 20 individual positive colonies from each transfection, then we tested expression levels of EGFP-tagged proteins by immunofluorescence. We selected cell lines expressing EGFP-fusion protein at a similar level to the endogenous protein, as measured by immunofluorescence stained kinetochores using Spc24 and Hec1-9G3 antibodies. Other EGFP strains were described in a previous study (*Suzuki et al., 2015*; *Suzuki et al., 2014*). All cells were monoclonal cell lines. RNAi experiments were conducted using LipofectamineRNAi MAX (Invitrogen) according to the manufacturer's instructions. siRNA was performed with 100 nM of siRNA duplex and siRNA sequences of CENP-T, CENP-C, Ndc80, and Nuf2 were described previously (*DeLuca et al., 2002*; *Suzuki et al., 2015*). Sequences for Spc24 (GGUCGACGAGGA-CACGACA), Spc25 (CUGCAAAUAUCCAGGAUCU), and Nuf2 (GAAGUCAUGUAUCCACAUU) were used. The efficiency for depletion of CENP-T, CENP-C, and Hec1 was greater than 95% by kinetochore immunofluorescence and western blot (*Suzuki et al., 2015*). The efficiency for Nuf2, Spc24, and Spc25 was 90, 80, and 70%, respectively, by kinetochore immunofluorescence using Hec1-9G3 antibodies. (We could not obtain antibodies that specifically recognized Nuf2, Spc24, or Spc25.) Although we used siRNA for endogenous proteins when we used cell lines expressing an EGFP tagged protein, there was no significant difference for Delta, 2D, and 3D separation measurements in Ndc80-EGFP stable cells with or without siRNA for Ndc80.

### Sample preparation

Cells were grown on acid washed #1.5 coverslips (~0.17 mm thick, Corning). For *Figure 5* experiments, cells were also grown on acid washed #1 or #2 coverslips (Corning). For samples embedded by PBS, cells were grown on 35 mm glass bottom dish with #1.5 coverslip (MatTek). Cells were fixed by pre-warmed 3% PFA (freshly made from paraformaldehyde powder (Sigma)) at 37°C in PHEM buffer (120 mM Pipes, 50 mM HEPES, 20 mM EGTA, 4 mM magnesium acetate, pH7.0) for 10–15 min for all experiments except for those in *Figure 7*. For *Figure 7* experiments, we fixed cells following nine different kinds of fixation (details in *Table 4*). Method 1: cells were fixed by 3% PFA in PHEM buffer for 10–15 min. Method 2: cells were fixed by 3% PFA in PHEM for 1 min, then cells were incubated in 0.5% triton in PHEM buffer, and cells were re-fixed by 3% PFA in PHEM buffer for an additional 10–15 min. Method 3: cells were incubated in 1% triton in PHEM for 1 min before fixation, then cells were fixed by 3% PFA in PHEM for 10–15 min. Method 4: cells were fixed by 1% GA (Glutaraldehyde, Sigma) in PHEM buffer for 10–15 min. Method 5: cells were fixed by 1% GA with 1% triton in PHEM buffer for 10–15 min. Method 6–7: cells were incubated in 1% triton before

fixation, then cells were fixed by 1% GA (Method 6) or 1% GA with 3% PFA (Method 7) in PHEM buffer for 10–15 min. Method 8: cells were fixed by cold MeOH for 15 min at −20 ℃. Method 9: cells were fixed by 3% PFA with 0.5% triton in PHEM buffer. Method 1 was used in the following studies (*Suzuki et al., 2014*; *Suzuki et al., 2011*; *Tauchman et al., 2015*; *Wan et al., 2009*), and this study). Methods 5 and 9 were used in *Magidson et al. (2016)*. All fixation, except for Method 8, was performed at 37 ℃. For samples fixed by GA (Method 4–7), after fixation, samples were incubated in NaBH$_4$ (Sigma) for quenching to reduce autofluorecence. Fixed samples were permeabilized by 0.5% NP40 (Roche) in PHEM buffer, and incubated in 0.5% BSA (Sigma) or BGS (Boiled Goat Serum) or BDS (Boiled Donkey Serum) for 30 min at room temperature. Then, samples were incubated for 1.5 hr at 37℃ with primary antibodies. Primary antibodies used in this study are listed in *Figure 5— source data 2*. CENP-A, CENP-T, and CENP-I antibodies were a gift from Drs. Aaron Straight, Iain Cheeseman, and Song-Tao Liu (*Carroll et al., 2009*; *Gascoigne et al., 2011*; *Liu et al., 2003*). After primary antibody incubation, samples were incubated for an additional 1.5 hr at 37℃ with secondary antibodies. Secondary antibodies were conjugated with Alexa488 or Rhodamine Red-X (Jackson ImmunoResearch). After DNA staining by DAPI, samples were mounted using Prolong Gold Aintifade (Molecular Probe) or PBS. For samples mounted with Prolong Gold, we usually waited >1 week for the refractive index to increase to near 1.46.

## Imaging

For image acquisition, 3D stacks of 70–90 frame pairs of red and green fluorescent images were obtained sequentially at 200 nm steps along the z axis through the cell using MetaMorph 7.8 software (Molecular Devices) and a high-resolution Nikon Ti inverted microscope equipped with an Orca AG cooled CCD camera with gain set to zero (Hamamatsu) and an 100X/1.4NA (PlanApo) DIC oil immersion objective (Nikon). Stage movement was controlled by MS2000-500 (ASI) with piezo-stage for z-axis stepping (*Suzuki et al., 2015*; *Suzuki et al., 2014*). The image magnification yielded a 64 nm pixel size. Solid state laser (Andor) illuminations at 488 and 568 nm were projected through Borealis (Andor) for uniform illumination of a spinning disk confocal head (Yokogawa CSU-10; Perkin Elmer). Raw 12 bit images without any image processing were used for all Delta, 2D, 3D separation analysis and signal intensity analysis.

## Intensity analysis

Integrated fluorescence intensity (minus local back ground (BG)) measurements were obtained for kinetochores as described previously (*Suzuki et al., 2015*). A 10 × 10 pixel region was centered on a fluorescent kinetochore to obtain integrated fluorescence, whereas a 14 × 14 pixel region centered on the 10 × 10 pixel region was used to obtain surrounding BG intensity. Measured values were calculated by: Fi (integrated fluorescence intensity minus BG)=integrated intensity for 10 × 10 region – (integrated intensity for the 14 × 14 – integrated intensity for 10 × 10) x pixel area of the 10 × 10/(pixel area of the 14 × 14 region – pixel area of a 10 × 10 region). Measurements were made with Metamorph 7.7 software (Molecular Devices) using Region Measurements (*Suzuki et al., 2015*). For *Figure 7*, the local kinetochore background signals were measured by (integrated intensity for the 14 × 14 – integrated intensity for 10 × 10) x pixel area of the 10 × 10/(pixel area of the 14 × 14 region – pixel area of a 10 × 10 region). At least two independent replicates of measurements were performed.

## Chromatic aberration (CA) calibration

To calibrate CA of our confocal microscope, we prepared samples with kinetochores stained by Hec1-9G3 primary antibodies and green (Alexa 488) and red (Rhodamine Red-X) labeled secondary antibodies. In addition, we also prepared coverslips with 100 nm or 500 nm TetraSpeck beads (Invitrogen) using a method described previously (*Churchman and Spudich, 2012*), except for using Prolong Gold as embedding media. X, Y, and Z coordinates of Green and Red fluorophores were determined by a 3D Gaussian fitting function in MatLab (MathWorks) (*Wan et al., 2009*). Details of CA correction methods are described in Text. Red and green fluorescent beads bound to the inner surface of a coverslip and embedded in Prolong Gold gave similar mean and SD values for CAx, CAy, CAz measured using Hec1-9G3-GR (Data not shown). At least two independent replicates of measurements were performed.

### Delta method

For each kinetochore, 3D centroid positions were first measured for each fluorescent color by a 3D Gaussian fitting function. For each sister kinetochore pair, the centroids of one color were projected to the axis defined by the centroids of the other color, and the average separation of the projected distance between the signals of different colors for that pair (*Wan et al., 2009*). At least two independent replicates of Delta measurements were performed.

### 2D and 3D separation measurements

For each single kinetochore, X, Y, Z coordinates for 2D or 3D centroid positions were measured for each fluorescent color by using a 3D Gaussian fitting function and saved in an Excel file. The values of X, Y, Z coordinates were copied into another Excel file named '2D 3D separation'. The 2D 3D separation file had all equations and calculated S(2D) and S(3D) using the equations described in Text and *Figure 1*. We applied Maximum likelihood fit (*Churchman et al., 2006*) for raw 2D or raw 3D values termed ML2D and ML3D when the separation was larger than 18 nm (See Text). The Matlab program called ML2D and ML3D read the raw 2D and raw 3D values in the Excel file and applied maximum likelihood fit to produce ML2D and ML3D. Then, the final ML2D and ML3D values with SD were copied from Matlab to the 2D 3D separation Excel file. The 2D 3D separation Excel file also provided the values of angle α and β (detail in Text). Note, because angles α and β were the angles relative to K-K axis, it required placing the x, y, z coordinates of sister kinetochore pairs in order. Then, the values were reorganized based on their x coordinate values within sister kinetochore pair in the Excel file. To obtain angle α and β values and also to directly compare the separation values by using Delta, 2D and 3D methods with same data sets, we usually measured sister kinetochore pairs in this study. However, the values for S(2D), S(3D), ML2D, and ML3D do not require sister kinetochore pairs. The programs used in this study can be obtained in the supplementary materials (*Source code 2–3*). At least two independent replicates of 2D and 3D separation measurements were performed.

### Statistical analysis

Statistical significance was determined using two tailed unpaired student's t-test for comparison between two independent groups. For significance, **$p < 0.05$ was considered statistically significant. The Delta, 2D, 3D separation and signal intensity experiments were replicated at least two independent experiments.

### Computational simulations

The computational simulation used for *Figures 8* and *9* was described in the text, *Figure 8*, and in the MatLab computer simulation files (*Source code 1–3*). The simulation code is available as a supplementary material.

## Acknowledgements

We thank Dr. Frederick Lanni for providing axial intensities of point spread functions based on his paper (*Gibson and Lanni, 1992*). We also thank Dr. Kerry Bloom and Josh Lawrimore for critical reading of the manuscript. This work was supported by R01GM24364 (EDS) from National Institute of Health.

## Additional information

### Funding

| Funder | Grant reference number | Author |
| --- | --- | --- |
| National Institutes of Health | R01GM24364 | Edward Salmon |

The funders had no role in study design, data collection and interpretation, or the decision to submit the work for publication.

## Author contributions
Aussie Suzuki, Conceptualization, Resources, Data curation, Software, Formal analysis, Funding acquisition, Investigation, Visualization, Methodology, Writing—original draft, Writing—review and editing; Sarah K Long, Data curation; Edward D Salmon, Conceptualization, Resources, Software, Supervision, Writing—original draft, Project administration, Writing—review and editing

## Author ORCIDs
Aussie Suzuki (iD) http://orcid.org/0000-0001-7390-5116

## Decision letter and Author response
Decision letter https://doi.org/10.7554/eLife.32418.036
Author response https://doi.org/10.7554/eLife.32418.037

## Additional files

### Supplementary files
• Source code 1. SimulationFluorCoLocal11282017Ssd Simulations used for *Figures 8–9*.
DOI: https://doi.org/10.7554/eLife.32418.031

• Source code 2. MLp2D Simulations used for obtaining ML2D values.
DOI: https://doi.org/10.7554/eLife.32418.032

• Source code 3. MLp3D Simulations used for obtaining ML3D values.
DOI: https://doi.org/10.7554/eLife.32418.033

• Transparent reporting form
DOI: https://doi.org/10.7554/eLife.32418.034

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
