## [Decision Letter]

[Editors’ note: a previous version of this study was rejected after peer review, but the authors submitted for reconsideration. The first decision letter after peer review is shown below.]

Thank you for submitting your work entitled "Protein Architecture of the Human Kinetochore/Microtubule Interphase at Metaphase from 3-D Fluorescence Co-Localization" for consideration by *eLife*. Your article has been reviewed by three peer reviewers, one of whom, Andrea Musacchio (Reviewer #1), is a member of our Board of Reviewing Editors, and the evaluation has been overseen by Vivek Malhotra as Senior.

Our decision has been reached after consultation between the reviewers. Based on these discussions and the individual reviews below, we regret to inform you that your work will not be considered further for publication in *eLife*.

As you will see, all three reviewers recognise the considerable merit of your work. They also recognise that previous work leading to the development of the "δ" concept in the Salmon laboratory has been transformative for the kinetochore field. Finally, the reviewers appreciate that the work presented here confirms the general usefulness of the method, and is therefore timely, also in consideration of the recent work from the Khodjakov and McAinsh laboratories, part of which published in *eLife*, which questioned some aspects of the work.

All this said, however, the three reviewers and the Reviewing Editor share the view that the current form of the manuscript lacks the clarity required to be accepted by the community as a generally useful technical resource. All three reviewers advocate an almost complete re-writing of the manuscript. As we realise that this will require considerable time and effort, we feel forced to return the manuscript to you and leave you an opportunity to decide on further moves. We will be happy to consider a revised form of the manuscript that fully meets the indications provided by the reviewers. This, however, will have to be considered a new submission.

Reviewer #1:

In this manuscript, the authors revisit and extend a method for 3D fluorescence co-localization that they originally developed with the goal of measuring distances between various proteins in the kinetochore. In short, the method allows assigning three-dimensional coordinates to the centroids of the fluorescence distributions of selected proteins (or antibodies that bind to them), and, after appropriate corrections, determining their mean distance. The layered distribution of proteins makes this approach particularly suited to the study of kinetochores. However, two recent papers (Smith et al., 2016; Magidson et al., 2016) have questioned, on different grounds, the conclusions of the authors' previous work, and solicited this response.

The Smith et al. paper applied a conceptually similar approach to the one originally proposed by the authors, but extended the analysis to three dimensions, finding co-localization distances that were significantly larger than those originally described by the authors. The Magidson et al. paper, on the other hand, questioned the foundation of the approach, in particular because it found, through an electron microscopy analysis, that the surface of kinetochores undergoes shrinking and becomes uneven under the fixation conditions originally used by the authors. Here, the authors appear to respond convincingly to both these previous papers. Their broad and thoughtful analysis identified all potential sources of errors in their methodology (the variance in fluorescence centroid determination, the variance of the mean chromatic aberration used for correction, and the thickness of the coverslip), and demonstrates that it is possible to obtain largely consistent distance measurements in one, two, and three dimensions provided that these sources of error are adequately addressed. Furthermore, the authors demonstrate that their approach provides estimates of distances that are largely independent of the specific fixation procedure.

Overall, the study provides further support to the usefulness of a method that has already contributed quite decisively to our understanding of kinetochore organization. Although the study is largely confirmatory for what regards kinetochore architecture (addressing which was not the main purpose of the manuscript), confirmation of the method's robustness has the potential to provide the community with a tool that can contribute to resolve crucial questions in kinetochore biology, in particular for what concerns the effects of force in kinetochore architecture.

On a less positive tone, as written the paper is a true heavy weight, and I rather strongly feel that the authors have to work very seriously to streamline their descriptions, simplify the figures, and shorten the text to make the paper more accessible. For instance, the mathematical analysis currently in Figure 3 could be summarized and presented in a separate box. Each figure comes with approximately 15 panels, and it would be preferable to present most of these data in form of a table in the main text and as plots in Supplementary. Also, the authors may want to harmonize font size throughout their figures, as the figures look very heterogeneous.

Reviewer #2:

Summary of the work and its relation to previous publications:

This paper presents exhaustive measurements of nanoscale distances between components within the kinetochores of human tissue-culture cells (HeLa cells). It is one of a series of studies, beginning with ground-breaking work from the Salmon lab (Wan et al., 2009), aimed at mapping the arrangement of kinetochore components, i.e. "the protein architecture of the kinetochore", in human cells fixed at metaphase of the cell cycle. The overall approach is to label by immunofluorescence two different kinetochore components with markers of two different colors, and then to measure the "intra-kinetochore distance" between the centroids of the two partially overlapping spots with sub-wavelength accuracy, from optical micrographs. The method is beautifully simple in principle. However, it is challenging in practice for technical reasons. There are limitations inherent to multi-color optical microscopy, such as the inevitable misregistration between images recorded in different color channels, and the statistical uncertainty in localization of any sub-wavelength object of finite brightness. There are also limitations due to prep-to-prep, cell-to-cell, and kinetochore-to-kinetochore variability. Nevertheless, sub-wavelength optical mapping of intra-kinetochore distances has provided very important information about how kinetochore components are arranged in cells, on average, and has had a major impact on how mitosis researchers view the kinetochore. A key concept has emerged, that changes in certain intra-kinetochore distances ('intra-kinetochore stretch') might provide the molecular basis for selective silencing of checkpoint signals and stabilization of microtubule attachments when sister kinetochores are properly bioriented.

This new paper is a response to recent publications from other groups that raised questions about the interpretation of sub-wavelength kinetochore maps (Magidson et al., 2016 and Smith, McAinsh and Burroughs, 2016). In earlier work, intra-kinetochore distances were measured as projections along a line, drawn between two sister kinetochores in the horizontal (x-y) plane, and then averaged to give a single value, called "Δ", for both sister kinetochores. The Δ method corrects locally for color mis-registration, because chromatic shifts will be opposite for kinetochores facing in exactly opposite (anti-parallel) directions. However, the Δ method ignores the possibility that two sister kinetochores might have different architectures or non-anti-parallel orientations. The studies of Magidson et al. and Smith et al. suggest that there can be significant tilting or swiveling of the outer portions of a kinetochore relative to the line connecting it to its sister, especially after treatment with microtubule poisons such as taxol or nocodazole. Magidson et al. also used correlative light and electron microscopy to show that individual kinetochores can become distorted, and that deviations from the canonical trilaminar plate morphology might correlate with high Δ values. Both Magidson et al. and Smith et al. suggest intra-kinetochore distances should be measured in two or three dimensions (2D or 3D), to account for tilting/swiveling, and both warn that projection onto the (1D) line connecting sisters, and averaging over the sisters, as in the Δ method, could underestimate distances and lead to misleading conclusions about the effects of microtubule drugs. Magidson et al. further show that different fixation methods (e.g., glutaraldehyde versus formaldehyde) can change the measured distances. Overall, these papers suggest exercising caution when interpreting Δ measurements.

In response, the new paper by Suzuki et al. presents an enormous amount of data. Control experiments with two colors labeling the same epitope are used to estimate chromatic shifts in 2D and 3D, for two different types of fixation and two different mounting media. Intra-kinetochore distances from the 9G3 epitope on Ndc80 to many other epitopes, on Cenp-T, Nuf2, Spc24, Spc25, and Cenp-A, are mapped using three different analysis methods, including the 1D Δ method, 2D, and fully 3D methods. Besides raw mean intra-kinetochore distances (+/- standard deviations), maximum likelihood methods are also applied in 2D and 3D, to account for the non-gaussian distributions expected whenever distances between two imperfectly localized objects are measured. The 9G3-to-Cenp-T distance is also mapped repeatedly for nine different fixation/permabilization protocols. Simulations were performed to assess how uncertainties in spot localization and in chromatic shifts influence the estimates of intra-kinetochore distance and tilt angles when using the various (2D and 3D) analysis methods. It is suggested that the true amount of kinetochore tilting is negligible, on the basis that a large fraction of the variability in measured tilt angles can be explained by uncertainties in localization and chromatic shift. The possibility of kinetochore-to-kinetochore variability in the measured distances is likewise discounted, again on the basis that much of the apparent variability can be explained by uncertainties in localization and chromatic shift. The possibility of cell-to-cell variability is discounted on the basis that the mean intra-kinetochore distance (between 9G3 and Spc25-EGFP) for a set of 120 individual kinetochores within a single cell is indistinguishable from that of 140 sister pairs in several cells. The overall conclusions are that kinetochore architecture is quite uniform, sisters are very nearly anti-parallel, and previous suggestions to the contrary were flawed because localization uncertainty was not adequately taken into account.

General assessment:

While the shear amount of data here is impressive, it unfortunately is not presented in an accessible way. Without a much clearer presentation, I do not feel the manuscript in its current form can serve as a useful technical resource. In many instances, it is extremely difficult to understand what is being plotted, and what are the primary important messages that the authors wish to draw from each piece of data. Some interesting and potentially valuable technical concepts are considered, such as the likelihood that variation in coverslip thickness causes large chromatic shifts in the z-direction, and the possibility that kinetochore tilting seen by other groups could be an artifact due to the inherent uncertainty in localizing dim fluorescence spots. However, it is extremely difficult to discern whether either of these potentially interesting points can be supported by the data. In my view, this paper is not ready for publication. Below I have listed specific comments which I hope can help the authors improve their presentation.

Substantive concerns:

Results section: "Because other Δ measurements had smaller SDs, we settled on values of Sxsd=Sysd = 4 and Szsd = 8, since there could have been some unexpected CA contribution to Δ variance from unknown local changes in refractive index." This seems arbitrary. The sentences just prior to this one suggested that larger values were needed to explain the data. Why go through the trouble to estimate carefully what is needed to explain the data, only to then impose an ad hoc reduction?

Results section: "The simulation data also shows that the combination of variances in centroid measurement and CA correction can account for a large fraction of the variance in tilt angles[…]This suggests that the majority of proteins linking the inner to outer kinetochore are aligned along the kMT axis." This is one of the key conclusions of the paper but it is not well supported. A thorough analysis would state explicitly how much of the variability is explained by the uncertainties in localization and chromatic shift, and also how much real kinetochore tilting would be consistent with the data.

Results section: "after depletion of the endogenous protein by RNAi" How effective was the depletion?

Results section: "This result indicates that most of the Z-offset in the original Sz data is not produced by unequal numbers of kinetochores from above and below the spindle axis, but likely from somewhat thinner coverslips than the standard 0.17 mm thickness (Figure 7)." This potentially interesting point is not rigorously supported without a demonstration that coverslips of known thickness but deviating from the standard 0.17 mm produce predictable changes in Z-offset. It seems that it would be a straightforward experiment to measure coverslips thicknesses and then prepare and analyze cells on those coverslips that happen to deviate from the standard.

Discussion section: "[…]we found that measurements of individual kinetochores (not sister pairs) within a single cell yielded the identical value at metaphase as the mean obtained from kinetochores within several cells. Thus, variations between cells or selected kinetochore measurements like Δ analysis is not an issue." This is overstated. The data in Figure 4 show only that mean values for populations of hundreds of kinetochores are consistent. But they say nothing about kinetochore-to-kinetochore variability. And "several cells" is unclear. How many cells? If the number is very small then the data cannot strongly support the conclusion that cell-to-cell variability is negligible.

The widespread, inconsistent, and redundant use of variables in this paper is extremely confusing. Figure 3 alone introduces at least 25 different variables: X1p, Y1p, Z1p, Sxsd, Sysd, Szsd, CAxp, CAyp, CAzp, CAxsd, CAysd, CAzsd, randn, Sxp, Syp, Szp, Sxyp, Sxyzp, CAxe, CAxs, CAye, CAys, CAze, CAzs. Sometimes two different variables seem to denote the same quantities, for example in Figure 1, where both CAx and Xg – Xr are used, whereas in Figure 1—figure supplement 1, only Xg – Xr is used. Conversely, two different calculations are sometimes represented by the same variable, such as S(3D), which does not include the correction for chromatic shift in Figure 1, but does include the correction in later figures. When the variables x and y are used in the equations for regression lines, they are often contradictory with respect to the plotted quantities, such as in Figure 1—figure supplement 1, where the regression variable x sometimes denotes Y (y-axis position in pixels) and the regression variable y sometimes denotes Xg-Xr (chromatic shift along x-axis in nanometers). When used sparingly and consistently, variables can make difficult mathematical concepts easier to follow. But here the use of variables seems to obfuscate things. It might help to adopt vector notation, to allow more compact expressions that capture the important conceptual relationships without explicitly showing all three cardinal directions.

The paper includes a huge number of scatterplots (I count ~59) and histograms (~32). Most of the scatterplots lack clear trends with respect to changes in the x-, y-, or z-position. In general, these plots seem to provide lots of clutter without adding much insight.

Reviewer #3:

This manuscript addresses a significant challenge: how to appropriately localize kinetochore components to measure distances between them. Different distance measurement methods have been used, and have at times led to different results. To move forward on solid ground, discrepancies must be reconciled, understood and explained. This is what this works attempts to do: it systematically and carefully explores the effect of different parameters (centroid determination, chromatic aberration correction, coverslip thickness and fixation method) on measured distances. The authors also provide a helpful computational framework for exploring how the variance in some of these parameters affects distance measurements. Importantly, the manuscript presents a 'molecular ruler' for checking the accuracy of 1D, 2D and 3D localization of kinetochore components by using the Ndc80 complex as a standard. Pending some changes outlined below, this work can become a significant contribution as a tool and resource, not just to those interested in kinetochores but to those interested in a broad array of cellular macromolecular assemblies.

Major concerns:

The paper as written is challenging to read and needs significant changes to both the text and figures to become an accessible – and used – resource.

1) While it is clear that the authors achieve more accurate centroid localization than previous works, the text does not clearly synthesize which practices together made this possible, and what experimental and analysis pipeline people should use to achieve such accuracy (what's good enough, and what isn't?). As one example to help guide changes, Figure 7 illustrates how Sz varies with imaging depth and kinetochore tilt, but the text describing this figure does not explain how this data relates to the rest of the work nor makes a concrete suggestion about best practices. Making a clear connection between the data and actionable outcomes is critical for this work to be a useful tool, and will help create a constructive, positive tone towards improving future measurements. On this note, it would be helpful to integrate in a single new figure the final suggested pipeline for obtaining accurate 3D measurements – unless the authors have a better idea for how to do that.

2) The figures are unnecessarily complicated and often very hard to read. The authors should take the time to think about which figures are essential for the main text and which are not, and should take the time to make their figures clear enough to stand alone. Figures should be significantly simplified. As examples, Figure 1 could be moved to the supplement and Figure 5 and Figure 6 could be moved or condensed. (In addition, it is critical that plots in all figures have clearly labeled axes in readable font sizes, and there must be no confusion about what is measured).

[Editors’ note: what now follows is the decision letter after the authors submitted for further consideration.]

Thank you for resubmitting your work entitled "An Optimized Method for 3D Fluorescence Co-Localization Applied to Human Kinetochore Protein Architecture" for further consideration at *eLife*. Your revised article has been favorably evaluated by Andrea Musacchio (Senior editor) and two reviewers.

This new version of the manuscript is greatly improved and we thank you for the effort you put into your resubmission. There are however some remaining issues that you might elect to address in a final version of the manuscript before acceptance, as outlined below. I have retained the reviewers' comments without editing them, as both reviewers feel that you should feel free to decide whether to incorporate change or not. I would be grateful if you could consider these points carefully, especially the major comments expressed by Reviewer 2.

Reviewer #1:

This is a resubmission of a previously reviewed manuscript. This manuscript addresses a significant challenge: how to appropriately localize kinetochore components to measure distances between them. Different distance measurement methods have been used, and have at times led to different results. To move forward on solid ground, discrepancies must be reconciled, understood and explained. This is what this works does, using both experiment and computation. The new manuscript largely addresses my two previous major concerns: it now synthesizes what experimental and analysis pipeline people should be used to achieve optimal accuracy, and presents simplified/clearer figures and text. This work is now a significant contribution as a tool and resource, not just to those interested in kinetochores but to those interested in a broad array of cellular macromolecular assemblies.

Reviewer #2:

This paper is very significantly improved. I commend the authors for their very careful reworking. I think the result is a paper that can serve as a very important resource, not only to the community of scientists interested in kinetochore architecture, but also more broadly, to microscopists studying the architectures of other large molecular assemblies inside cells. I hope the following relatively minor comments will help the authors further improve their manuscript in preparation for publication. In particular, I feel the impact of the paper could be boosted by a more quantitative analysis of the amount of true, kinetochore-to-kinetochore variability that would be compatible with the measurements, as explained in the last two comments below.

Results section: "Unexpectedly, in vivo, the C-terminus of Ndc80 locates much closer to the globular domains of Spc24/Spc25 (Figure 4). This difference might be caused by structural restrictions on the position of the EGFP tag, which was tethered to the protein end by a 9 amino acid linker." A structure for the tetramerization domain within the Ndc80 complex is available (Valverde, Ingram and Harrison, 2016). Can the apparent arrangement in vivo of the C-termini of Ndc80 and Nuf2 and the N-termini of Spc24 and Spc25 be reconciled with this published structure? Or does one need to further invoke "restrictions" on the epitopes/antibodies to reconcile these different views of the Ndc80 complex? More generally, the published structures for the Ndc80 complex can provide predictions for the distances between many of the antibody pairs used in this study. Why not include a comprehensive, graphical comparison of predicted versus measured distances? If the correlation is good, then such a comparison would provide very compelling additional support for the accuracy of the fluorescence colocalization technique. It would also help clarify what is the in vivo structure of the Ndc80 complex.

Discussion section: "Note that their higher SDc values explains why their Δ measurements are significantly larger than ours (Figure 9—figure supplement 1)." This statement seems a bit overstated to me considering that the authors can only estimate the SDc values of their competitor's work – they do not know the SDc values from that work with 100% certainty. In my view, it would be more correct, and would not undercut the impact of the present paper at all, to qualify the sentence very slightly as follows: "Such high SDc values could explain[…]" or "Such high SDc values probably explain[…]"

Discussion section; "A previous study has shown that the accuracy of separation measurements is sensitive to the spatial staggering of the labeled molecules along the kinetochore inner to outer axis only for stagger greater than 150 nm, which produces two fluorescent peaks instead of a single peak (Joglekar et al., 2009)." This statement seems over-simplified to me, and potentially confusing. The production of two peaks is not a general consequence of staggering of the labeled molecules. It occurred in a highly idealized simulation in the cited paper, where the simulated positions of six red spots were shifted by fixed distances in the x-direction relative to a corresponding set of six green spots. More random staggered arrangements could retain a single fluorescent peak, but with a larger width compared to a case without staggering.

Discussion section: "Second, the SD of our 2D/3D measurements are small, typically less than 10 nm, and a significant part of this SD must be from the SDc and CAsd (see above discussion) and not from differences between kinetochores." This is interesting, but it only weakly supports the "uniform architecture" hypothesis. A key question is, how much true, kinetochore-to-kinetochore variability in distance is compatible with the measured SDs? Based on the data and analyses presented in this paper, it should be possible to make a quantitative statement about how wide the underlying distribution of true distances is likely to be. The purely qualitative statement, that the architecture is "very" uniform, seems a bit hollow.

Discussion section: "Surprisingly, only 10~15 kinetochores needed to be averaged to achieve the mean and SD typical of > 150 kinetochores (Figure 9—figure supplement 2)." This observation merely indicates that 10 to 15 kinetochores are sufficient to estimate the mean and width of the distribution. It does not by itself indicate anything about the width, i.e. about the level of kinetochore-to-kinetochore variability.

---

## [Author Response]

[Editors’ note: the author responses to the first round of peer review follow.]

All this said, however, the three reviewers and the Reviewing Editor share the view that the current form of the manuscript lacks the clarity required to be accepted by the community as a generally useful technical resource. All three reviewers advocate an almost complete re-writing of the manuscript. As we realise that this will require considerable time and effort, we feel forced to return the manuscript to you and leave you an opportunity to decide on further moves. We will be happy to consider a revised form of the manuscript that fully meets the indications provided by the reviewers. This, however, will have to be considered a new submission.

We appreciate that all reviewers found our study important and with considerable merit. We also thank the reviewers for their constructive comments and suggestions. As requested by the reviewers and the editor, both the text and figures have been reorganized and edited to “improve clarity of the manuscript to the level required by the community as a general useful technical resource.”

Significant modifications in the revised manuscript to improve clarity are as follows:

1) Addition of a “Principles of the Method” section to the beginning of the Results with a corresponding Figure containing schematic and numerical information of the analytical method to provide the requested “pipeline for obtaining accurate 3D measurements”, and Figure 1—source data 1 providing details of each step and potential error sources.

2) Re-organization of all figures to put key findings in the main Figures and supporting data in corresponding Figure supplement in a logical way to comply with their presentation in *eLife*.

3) Unified the names of variables used in both the experimental data analysis and the computer simulation of the method to make the fundamental principles clearer for the reader to follow.

4) Provided new data in the Materials and methods to address the question from the reviewers about missing data: the efficiency of endogenous protein depletion by siRNA for the EGFP labeled protein components of the Ndc80 complex.

5) Made modifications or additions to the text in response to the specific reviewer’s concerns as listed below.

Reviewer #1:[…]On a less positive tone, as written the paper is a true heavy weight, and I rather strongly feel that the authors have to work very seriously to streamline their descriptions, simplify the figures, and shorten the text to make the paper more accessible. For instance, the mathematical analysis currently in Figure 3 could be summarized and presented in a separate box. Each figure comes with approximately 15 panels, and it would be preferable to present most of these data in form of a table in the main text and as plots in Supplementary. Also, the authors may want to harmonize font size throughout their figures, as the figures look very heterogeneous.

We addressed this issue as summarized in “Significant modifications in the revised manuscript to improve clarity”.

Reviewer #2:[…]General assessment:While the shear amount of data here is impressive, it unfortunately is not presented in an accessible way. Without a much clearer presentation, I do not feel the manuscript in its current form can serve as a useful technical resource. In many instances, it is extremely difficult to understand what is being plotted, and what are the primary important messages that the authors wish to draw from each piece of data. Some interesting and potentially valuable technical concepts are considered, such as the likelihood that variation in coverslip thickness causes large chromatic shifts in the z-direction, and the possibility that kinetochore tilting seen by other groups could be an artifact due to the inherent uncertainty in localizing dim fluorescence spots. However, it is extremely difficult to discern whether either of these potentially interesting points can be supported by the data. In my view, this paper is not ready for publication. Below I have listed specific comments which I hope can help the authors improve their presentation.

We addressed the general “clarity” issue as summarized in “Significant modifications in revised manuscript to improve clarity”.

Substantive concerns:Results section: "Because other Δ measurements had smaller SDs, we settled on values of Sxsd=Sysd = 4 and Szsd = 8, since there could have been some unexpected CA contribution to Δ variance from unknown local changes in refractive index." This seems arbitrary. The sentences just prior to this one suggested that larger values were needed to explain the data. Why go through the trouble to estimate carefully what is needed to explain the data, only to then impose an ad hoc reduction?

To clarify this issue, we edited the relevant paragraph in the results to read (Note Ssd is now named SDc for variance in centroid determination):

“To obtain an estimate of the error contributed from the SDc, we used results from Δ analysis which in principle have local CA correction. […] This result indicates that the variance in our centroid measurement is very small and only slightly enhances the 3D measurements from the true value.”

Results section: "The simulation data also shows that the combination of variances in centroid measurement and CA correction can account for a large fraction of the variance in tilt angles[…]This suggests that the majority of proteins linking the inner to outer kinetochore are aligned along the kMT axis." This is one of the key conclusions of the paper but it is not well supported. A thorough analysis would state explicitly how much of the variability is explained by the uncertainties in localization and chromatic shift, and also how much real kinetochore tilting would be consistent with the data.

We deleted this conclusion from the simulation section of the revised manuscript and inserted the following sentence at the end of the section of the Results entitled: “The Mean Human Kinetochore Protein Architecture at Metaphase Measured by the Δ Method and Our 3D Method are Similar.” to read:

“In Figure 5, the Ndc80 complex and protein linkage to the inner kinetochore are shown aligned with the axis of kMTs because the ML3D mean separation values are only slightly greater than their corresponding Δ values and the Δ value is a projection of the ML3D value along the K-K axis.”

In addition, we have added the potential errors in angle measurements in the result section of kinetochore tilt and simulation in revised manuscript to read:

“Unlike 3D separation, the mean tilt angle β was sensitive to unequal numbers of kinetochores above and below the spindle equator. The mean angle β is near zero in the middle of the spindle and reaches mean values of about 20-25 degrees at the bottom and top of the spindle (Figure 6—figure supplement 2).”

and

“To see how the SDc and CAsd of our measurement method effect the variances in the angles α and β, we entered into the simulation constant values for S between zero and 100 nm (Figure 8). The variance in the angles α and β are largest for small separations (Figure 8). Our SDc and CAsd caused about half to two-thirds of the variance measured in cells for separations of 60 nm (CENP-T-Hec1-9G3 in Figure 3) or 90 nm (CENP-A-Hec1-9G3 in Figure 6—figure supplement 2).”

Results section: "after depletion of the endogenous protein by RNAi" How effective was the depletion?

We have added the efficiency of protein depletion in Materials and methods of the revised manuscript. The new sentences read:

“siRNA was performed with 100 nM of siRNA duplex and siRNA sequences of CENP-T, CENP-C, Ndc80, and Nuf2 were described previously (DeLuca et al., 2002; Suzuki et al., 2015). […]Although we performed siRNA for endogenous proteins when we used cell lines expressing EGFP tagged protein, there was no significant differences for Δ, 2D, and 3D separation measurements in Ndc80-EGFP stable cells with or without siRNA for Ndc80.”

Results section: "This result indicates that most of the Z-offset in the original Sz data is not produced by unequal numbers of kinetochores from above and below the spindle axis, but likely from somewhat thinner coverslips than the standard 0.17 mm thickness (Figure 7)." This potentially interesting point is not rigorously supported without a demonstration that coverslips of known thickness but deviating from the standard 0.17 mm produce predictable changes in Z-offset. It seems that it would be a straightforward experiment to measure coverslips thicknesses and then prepare and analyze cells on those coverslips that happen to deviate from the standard.

To address this issue, we added a new Graph and the following sentence:

“The plot in Figure 6—figure supplement 1 illustrates the sensitivity of Z-offset to coverslip thickness. A steep slope is seen when measured Z-offset values in Figure 6 are plotted as a function of the mean thickness we measured for #1, #1.5 and #2 coverslips.”

Discussion section: "[…]we found that measurements of individual kinetochores (not sister pairs) within a single cell yielded the identical value at metaphase as the mean obtained from kinetochores within several cells. Thus, variations between cells or selected kinetochore measurements like Δ analysis is not an issue." This is overstated. The data in Figure 4 show only that mean values for populations of hundreds of kinetochores are consistent. But they say nothing about kinetochore-to-kinetochore variability. And "several cells" is unclear. How many cells? If the number is very small then the data cannot strongly support the conclusion that cell-to-cell variability is negligible.

The above sentence was miss-worded. The corrected sentences read:

“Sister kinetochore pairs were used for majority of measurements in this study to directly compare separations obtained by Δ and 2D/3D fluorescent co-localization methods using the same data sets. To test if selection of sister pairs influenced our measurements, we measured 120 kinetochores selected randomly in a single cell. This produced a mean separation value nearly identical to the mean value for single kinetochore measurements of sister pairs obtained from 6 metaphase cells (Figure 4).”

The widespread, inconsistent, and redundant use of variables in this paper is extremely confusing. Figure 3 alone introduces at least 25 different variables: X1p, Y1p, Z1p, Sxsd, Sysd, Szsd, CAxp, CAyp, CAzp, CAxsd, CAysd, CAzsd, randn, Sxp, Syp, Szp, Sxyp, Sxyzp, CAxe, CAxs, CAye, CAys, CAze, CAzs. Sometimes two different variables seem to denote the same quantities, for example in Figure 1, where both CAx and Xg – Xr are used, whereas in Figure 1—figure supplement 1, only Xg – Xr is used. Conversely, two different calculations are sometimes represented by the same variable, such as S(3D), which does not include the correction for chromatic shift in Figure 1, but does include the correction in later figures. When the variables x and y are used in the equations for regression lines, they are often contradictory with respect to the plotted quantities, such as in Figure 1—figure supplement 1, where the regression variable x sometimes denotes Y (y-axis position in pixels) and the regression variable y sometimes denotes Xg-Xr (chromatic shift along x-axis in nanometers). When used sparingly and consistently, variables can make difficult mathematical concepts easier to follow. But here the use of variables seems to obfuscate things. It might help to adopt vector notation, to allow more compact expressions that capture the important conceptual relationships without explicitly showing all three cardinal directions.

We significantly unified variable names between those used for experimental analysis and those used in the computer simulations.

The paper includes a huge number of scatterplots (I count ~59) and histograms (~32). Most of the scatterplots lack clear trends with respect to changes in the x-, y-, or z-position. In general, these plots seem to provide lots of clutter without adding much insight.

We have significantly modified Figure construction to put key data in the major figures and supporting data in Supplemental Figures using the *eLife* format.

Reviewer #3:Major concerns:The paper as written is challenging to read and needs significant changes to both the text and figures to become an accessible and used resource.1) While it is clear that the authors achieve more accurate centroid localization than previous works, the text does not clearly synthesize which practices together made this possible, and what experimental and analysis pipeline people should use to achieve such accuracy (what's good enough, and what isn't?). As one example to help guide changes, Figure 7 illustrates how Sz varies with imaging depth and kinetochore tilt, but the text describing this figure does not explain how this data relates to the rest of the work nor makes a concrete suggestion about best practices. Making a clear connection between the data and actionable outcomes is critical for this work to be a useful tool, and will help create a constructive, positive tone towards improving future measurements. On this note, it would be helpful to integrate in a single new figure the final suggested pipeline for obtaining accurate 3D measurements – unless the authors have a better idea for how to do that.

We have significantly modified the manuscript and figures to address these concerns as outlined in the first page in “Significant modifications in revised manuscript to improve clarity.”

2) The figures are unnecessarily complicated and often very hard to read. The authors should take the time to think about which figures are essential for the main text and which are not, and should take the time to make their figures clear enough to stand alone. Figures should be significantly simplified. As examples, Figure 1 could be moved to the supplement and Figure 5 and Figure 6 could be moved or condensed. (In addition, it is critical that plots in all figures have clearly labeled axes in readable font sizes, and there must be no confusion about what is measured).

Again, we have significantly modified the manuscript and figures to address these concerns as outlined in the first page in “Significant modifications in revised manuscript to improve clarity.”

[Editors' note: the author responses to the re-review follow.]

Reviewer #2:This paper is very significantly improved. I commend the authors for their very careful reworking. I think the result is a paper that can serve as a very important resource, not only to the community of scientists interested in kinetochore architecture, but also more broadly, to microscopists studying the architectures of other large molecular assemblies inside cells. I hope the following relatively minor comments will help the authors further improve their manuscript in preparation for publication. In particular, I feel the impact of the paper could be boosted by a more quantitative analysis of the amount of true, kinetochore-to-kinetochore variability that would be compatible with the measurements, as explained in the last two comments below.Results section: "Unexpectedly, in vivo, the C-terminus of Ndc80 locates much closer to the globular domains of Spc24/Spc25 (Figure 4). This difference might be caused by structural restrictions on the position of the EGFP tag, which was tethered to the protein end by a 9 amino acid linker." A structure for the tetramerization domain within the Ndc80 complex is available (Valverde et al., 2016). Can the apparent arrangement in vivo of the C-termini of Ndc80 and Nuf2 and the N-termini of Spc24 and Spc25 be reconciled with this published structure? Or does one need to further invoke "restrictions" on the epitopes/antibodies to reconcile these different views of the Ndc80 complex? More generally, the published structures for the Ndc80 complex can provide predictions for the distances between many of the antibody pairs used in this study. Why not include a comprehensive, graphical comparison of predicted versus measured distances? If the correlation is good, then such a comparison would provide very compelling additional support for the accuracy of the fluorescence colocalization technique. It would also help clarify what is the in vivo structure of the Ndc80 complex.

We have added a detailed comparison between our measurements and previous studies in vitro (Huis In 't Veld et al., 2016; Valverde et al., 2016; Wei et al., 2005) in Figure 4—figure supplement 1.

Discussion section: "Note that their higher SDc values explains why their Δ measurements are significantly larger than ours (Figure 9—figure supplement 1)." This statement seems a bit overstated to me considering that the authors can only estimate the SDc values of their competitor's work – they do not know the SDc values from that work with 100% certainty. In my view, it would be more correct, and would not undercut the impact of the present paper at all, to qualify the sentence very slightly as follows: "Such high SDc values could explain[…]" or "Such high SDc values probably explain[…]"

Thank you for an important suggestion. We have modified this and subsequent sentences to read;

“Such high CDsd values could explain why their Δ measurements are significantly larger than ours (Figure 9—source data 1). […]In our measurements, large values of Zoffset were rare (Figure 5—source data 1); we have no information to judge how much other sources of error like Z-offset, contributed to the Smith, McAinsh and Burroughs, 2016 measurements.”

Discussion section; "A previous study has shown that the accuracy of separation measurements is sensitive to the spatial staggering of the labeled molecules along the kinetochore inner to outer axis only for stagger greater than 150 nm, which produces two fluorescent peaks instead of a single peak (Joglekar et al., 2009)." This statement seems over-simplified to me, and potentially confusing. The production of two peaks is not a general consequence of staggering of the labeled molecules. It occurred in a highly idealized simulation in the cited paper, where the simulated positions of six red spots were shifted by fixed distances in the x-direction relative to a corresponding set of six green spots. More random staggered arrangements could retain a single fluorescent peak, but with a larger width compared to a case without staggering.

Our writing was not clear. This paragraph now reads:

“Human kinetochores are built from multiple copies of core-kinetochore proteins (Suzuki et al., 2016). […] This indicates that our centroid measurements correspond to the mean position of the fluorescently labeled protein epitopes within human kinetochores.”

Discussion section: "Second, the SD of our 2D/3D measurements are small, typically less than 10 nm, and a significant part of this SD must be from the SDc and CAsd (see above discussion) and not from differences between kinetochores." This is interesting, but it only weakly supports the "uniform architecture" hypothesis. A key question is, how much true, kinetochore-to-kinetochore variability in distance is compatible with the measured SDs? Based on the data and analyses presented in this paper, it should be possible to make a quantitative statement about how wide the underlying distribution of true distances is likely to be. The purely qualitative statement, that the architecture is "very" uniform, seems a bit hollow.Discussion section: "Surprisingly, only 10~15 kinetochores needed to be averaged to achieve the mean and SD typical of > 150 kinetochores (Figure 9—figure supplement 2)." This observation merely indicates that 10 to 15 kinetochores are sufficient to estimate the mean and width of the distribution. It does not by itself indicate anything about the width, i.e. about the level of kinetochore-to-kinetochore variability.

We asked how much kinetochore-to-kinetochore variability can be involved in our experimental measurements using simulations in Figure 9—source data 2. Based on these results, we have modified sentences to read: “Second, as expected for the Gaussian distribution measured for separation values (e.g. Figure 3), we found that only 10~15 kinetochores were needed to be averaged to achieve the mean and SD typical of > 150 kinetochores (Figure 9—figure supplement 1). Third, the SD of our 2D/3D measurements are small, typically less than 12 nm, and a significant part of this SD must be from the CDsd and CAsd (Table 6 and see above discussion) and not from differences between kinetochores. To see this more clearly, we used the same simulations in Figure 8–Figure 9 with mean kinetochore-to-kinetochore variability (Ksd = ± 0, ± 5, ± 10, or ± 15 nm) (Figure 9—source data 2). Simulation results showed that Ksd of about ± 5 nm or less plus measurement CDsd and CAsd yielded the experimental mean SD value of 12.5 nm (Figure 9—source data 2). These results imply that the vast majority of kinetochores at metaphase have a similar protein architecture”